

# Intermediate water flows in the South West Pacific from OUTPACE and THOT Argo floats

Simon Barbot[1], Anne Petrenko[1], and Christophe Maes[2]

[1]Aix Marseille Université , CNRS/INSU, IRD, Université du Sud Toulon-Var, Mediterranean Institute of Oceanography (MIO), UM 110, 13288 Marseille
[2]Laboratoire d'Océanographie Physique et Spatiale (LOPS), CNRS, Ifremer, IRD, UBO, Brest, France

*Correspondence to:* Simon Barbot (simon.barbot@legos.obs-mip.fr)

**Abstract.** Thanks to the autonomous Argo floats of the OUTPACE cruise and of the THOT project, some features of intermediate flow dynamics, around 1000 m depth, within the Southwest and Central Pacific Ocean ($156^o$E-$150^o$W, around $19^o$S) are described. In the Coral sea, we highlight minima in dissolved oxygen of 140 $\mu$mol.kg$^{-1}$ that are associated with the signature

of a southward transport of waters between two zonal jets: from the North Vanuatu Jet to the North Caledonia Jet. This transport takes place in the core of a cyclonic eddy or via the path between a cyclonic eddy and an anticyclonic one, highlighting the importance of mesoscale dynamics in upper thermocline and surface layers. Further east, we observe a strong meridional velocity shear with long-term float trajectories going either eastward or westward in the lower thermocline. More interestingly, these trajectories also exhibit some oscillatory features. Those trajectories can be explained by a single Rossby wave of 160 days

period and 855 km wavelength. Considering the thermohaline context, we confirm the meridional shear of zonal velocity and highlight a permanent density front that corresponds to the interface between Antarctic Intermediate Waters and North Pacific Deep Waters. Hence both circulation and thermohaline contexts are highly favorable to instabilities and wave propagation.

## 1  Introduction

The intermediate water masses have been known to be of great importance for the thermohaline overturning circulation

(Las Heras and Schlitzer, 1999) and for biogeochemical cycles (Ganachaud and Wunsch, 2002) all over the oceans. More specifically, the Western Tropical South Pacific (WTSP) interests the biogeochemistry community because of its oligotrophic and ultra-oligotrophic zones where diazotrophy has a large influence on phytoplankton growth (Bonnet et al., 2017; Benavides et al., 2017; Caffin et al., 2017). To make progress on this subject, the OUTPACE cruise (Oligotrophy to UlTra-oligotrophy PACific Experiment ; Moutin et al., 2017), on board the R/V L'Atalante, took place during spring 2015. The surface circulation

of the cruise context has been studied in order to understand the impact of meso- and submeso-scale on an observed bloom (de Verneil et al., 2017b), on the horizontal distribution of biogeochemical/biological components (Rousselet et al., 2017) and replacing them in the large-scale circulation of the Western Tropical South Pacific (Fumenia et al., 2018).

Our study will complete these observations of the surface circulation both in space, by focusing on intermediate levels, and in time, by using the time series (more than two years) of the autonomous Argo floats deployed during the cruise.



Autonomous Argo floats are useful to study intermediate circulation and biogeochemistry and biology of the first thousand meters of the water column, when they are also BGC floats. Such approaches lead to horizontal gridded maps of velocities either global (Davis, 2005; Ollitrault and Rannou, 2013) or regional (Cravatte et al., 2012). Other approaches are developed in order to study the intermediate circulation by taking into account the spreading of the different float trajectories for a same

position (Sevellec et al., 2017). In a sense, the studies focusing on the specificity of one particular trajectory (*e.i.* Mitchell, 2003; Boss et al., 2008; IOCCG, 2009; Bishop and Wood, 2009; Phillips and Bindoff, 2014) are less frequent now that data from many floats have become available. Our approach focuses on a few float trajectories using them as witnesses of intermediate waters specific dynamics.

The cruise started in the Coral Sea, where the main upper currents are characterized by narrow jets named the North Vanuatu

Jet (NVJ) and the North Caledonia Jet (NCJ) at the entrance of the Coral Sea. Both of them are westward zonal jets (Fig. 1a, Webb, 2000) but the water masses they carry at the level of the main thermocline are not identical. Gasparin et al. (2014) examine the composition of water masses and report a net difference in dissolved oxygen concentration (DOXY) with waters more deoxygenated in the NVJ than in the NCJ. These two jets derive from different branches of the South Equatorial Current (SEC), a broad current with prevailing geostrophic westward currents (Fig. 1a). The NVJ originates from a longer branch, hence

its lower DOXY. Using DOXY as a proxy to make the difference between NVJ and NCJ, Rousselet et al. (2016) revealed an intrusion of NVJ waters in the NCJ pathway during the BIFURCATION cruise (Maes, 2012). The former authors concluded that the NVJ waters were transported southward by a mesoscale anticyclonic eddy. Here the BGC Argo float deployed at the Long Duration station A (LDA) of OUTPACE, north of New Caledonia, has followed the path of the NCJ. Its BGC data allow to test whether, during its path, it has encountered water masses coming from the NVJ.

Further east, other Argo floats, deployed during OUTPACE, are characterized by a main eastward displacement in opposite direction compared to the westward SEC (Fig. 1a). Observations in other regions of the world revealed a complex alternation of zonal currents, now known as striations, in surface zonal velocity estimated with altimetric products (*e.g.* Maximenko et al., 2005, 2008). These striations have also been found, more recently, at intermediate levels with Argo floats (Fig. 1c, Cravatte et al., 2012; Ollitrault and Colin de Verdière, 2014). More detailed works are underway on such striations in order to observe

them more precisely and to understand their physics (Cravatte et al., 2017; Belmadani et al., 2017). During the OUTPACE cruise, Rousselet et al. (2017) highlight the velocity anomaly of such striations at the surface in one specific area close to the OUTPACE domain ($160^{o}$E-$150^{o}$W and $18^{o}$S-$22^{o}$S). Thanks to the floats we can extend this observation at intermediate levels and several years after the cruise. In our case, two intermediate water masses are present near the parking depth of the floats, around 1000 m: the Antarctic Intermediate Waters (AAIW) to the south and the North Pacific Deep Water (NPDW) to the north

(Fig. 1b).

The first part of this study focuses on the mesoscale interactions between NCJ and NVJ within the Coral Sea. Thanks to Argo float WMO6901656, equipped with an optode sensor, we analyze the DOXY variability over the 2015-2016 period using it as a proxy in order to distinguish NCJ waters from NVJ ones. Then, we replace the DOXY anomalies in their thermohaline and circulation context. In a second part, we focus on the central Pacific Ocean zone where we describe specific trajectories of

floats deployed during the OUTPACE cruise or in the framework of the THOT project (TaHitian Ocean Time series, Martinez



**Figure 1.** South-west Pacific Ocean (a) main trajectories of zonal jets superimposed on the bottom topography (in m), (b) absolute salinity (g.kg$^{-1}$) at 1000 m for November climatology from the ISAS13 atlas and (c) zonal mean current (in cm/s, positive for eastward velocity) deduced from the Argo float displacement at their parking depth (near 1000 m, adapted from Ollitrault and Colin de Verdière, 2014). The colored lines represent the studied floats (WMO number in the inset) with squares for their immersion location. The currents shown in the upper panel are the South Equatorial Current (SEC), the North Vanuatu Jet (NVJ), the North Caledonia Jet (NCJ), the North Queensland Current (NQC) and the East Australia Current (EAC). The water masses shown on the middle panel are the North Pacific Deep Water (NPDW) and the Antarctic Intermediate Waters (AAIW). The black line represents the shiptrack of the OUTPACE cruise and, in the middle panel, the crosses represent the short time stations (numbers) and the squares A and C represent two the long time stations of the cruise where the studied float were deployed.



et al., 2015). Once again, we also replace the float trajectories in their thermohaline and circulation context. In order to explain their displacement at mid-depths we choose a wave approach and compare our results, taking into account Doppler shift, to different cases of Rossby, Kelvin and Kelvin-Helmholtz instability waves. Then, we discuss the proposed hypothesis for the two areas and what they imply for the OUTPACE observations. Finally, we conclude and propose some method improvements.

## 2 Data sets and methodology

### 2.1 Autonomous floats

In the present study we benefit from the deployment during the 2015 spring of several Argo floats under the auspices of the OUTPACE cruise (Moutin et al., 2017) and of the THOT project (Martinez et al., 2015). We analyze the two first years of data. All OUTPACE and THOT floats are related to the Argo international program and have the same type of sampling cycle. For memory, it begins with the descent of the float to a depth around 1000 m, called the parking depth where the floats drift for a programmed time. This time can be remotely modified in the last generation of floats, like those used in this study. Here we use the measurements made while the floats rise to the surface. At the surface, these data are transmitted via satellite. Even if this cycle is the same for all floats, the time spent at the parking depth (in our cases, 5 or 10 days), and hence the corresponding sampling frequency ($\Delta t$), are not the same for all of them. Two types of float are available: ARVOR (Argo-Core) floats and PROVBIO floats (Bio-Argo). Because they were all immersed during the same months, they all have a close WMO code : #6901XXX. Hereafter, to ease the writing of the float number, we only use the three last digits to point to them, *i.e.* float 656 refers to #6901656. Details and data are accessible via the CORIOLIS operational center (www.coriolis.eu.org). In order to get the average velocity of the floats during their cycles, we follow the method of Ollitrault and Rannou (2013) taking different time slots over the float cycle in order to calculate, for the $n^{\text{th}}$ cycle, the surface velocity $V^n(0)$ and the velocity at 1000 m $V^n(1000)$ :

$$V^n(0) = \frac{L^n_{\text{last}} - L^n_{\text{first}}}{t^n_{\text{last}} - t^n_{\text{first}}} , \qquad V^n(1000) = \frac{L^n_{\text{first}} - L^{n-1}_{\text{last}}}{t^n_{\text{first}} - t^{n-1}_{\text{last}}} \qquad (1)$$

where $L_{\text{first}}$ and $L_{\text{last}}$ are the first and the last transmitted location in the same cycle (PROVBIO floats transmit only two locations per cycle using Iridium whereas ARVOR floats transmit seven locations per cycle using ARGOS) and $t$ is the corresponding time. After some verifications we have concluded that, even if the mean surface velocity is ten times greater than the mean deep velocity, each float stays such a short time at the surface that its displacement there can be neglected compared to the deep displacement. Hence we can consider that the trajectory dynamics are mainly due to deep circulation processes. The discussion of such consideration is thoroughly made by Ollitrault and Rannou (2013).

In the case of the 656 float, we also consider the dissolved oxygen concentration (DOXY) measurements. These data were calibrated (A. Fumenia, personal communication) based on the CTD profiles made just before the float was immersed during the OUTPACE cruise. The calibration for the 656 float is:

$$DOXY_{\text{clb}} = 0.97125 \, DOXY_{\text{raw}} + 14.3755 \qquad (2)$$





where $DOXY_\mathrm{raw}$ is the dissolved oxygen concentration data from the float and $DOXY_\mathrm{clb}$ is the calibrated dissolved oxygen concentration data, both expressed in $\mu\mathrm{mol.kg}^{-1}$ .

## 2.2   Trajectory description for the a wave approach

Here the objective is to explain the float trajectories influenced by waves. Thus, we describe the float trajectories as Lagrangian

description of waves, using the period $T$ and the wavelength $\lambda$ as well as the frequency $\omega$ ($\omega = 2\pi/T$) and the wavenumber $k$ ($k = 2\pi/\lambda$). The wavenumber is defined as:

$$k^2 = k_\mathrm{v}^2 + k_\mathrm{H}^2 = k_\mathrm{v}^2 + k_\mathrm{lon}^2 + k_\mathrm{lat}^2 \tag{3}$$

where $k_\mathrm{v}$ is the vertical wavenumber, $k_\mathrm{H}$ the horizontal one, $k_\mathrm{lon}$ the longitudinal one and $k_\mathrm{lat}$ the latitudinal one. In our case, the dynamics of the floats do not allow us to raise information concerning the vertical component because data on float locations

are only measured at the surface. The displacements of the floats are mostly directed by the currents, and only secondarily by waves. Since the prevailing currents are mainly zonal, we consider hereafter, to simplify, that the wavenumbers derived from the observations correspond to the longitudinal component ($k = k_\mathrm{lon}$). To clarify the methodology, we choose to name "float wave" the measured wavy trajectory of a float and "theoretical wave" the process that could lead to such trajectory.

We tried to describe the float waves with a Fourier transform or a wavelet analysis on the float time series, but the description

of the different frequencies contained in them was incomplete. This is due to the shortness of the time series (2 years) with regard to the sampling period. If the floats are still functional in a few years, these methods should be reconsidered.

So instead, we developed the method presented here, with Lagrangian and Eulerian wave descriptions and the estimations of wavenumbers. In order to maximize the information derived from the portions of the trajectories where the floats oscillate, we choose to determine half float wave characteristics rather than full float wave ones, for each float studied (floats 660, 671,

679 and 687, Fig.2). Hence, we obtain a double number of more precise measurements of $\lambda/2$ from the position maps, and of $T/2$ from the float time series of latitude.

Because the floats are, by definition, Lagrangian devices, we need to be careful before comparing the float waves to the classical Eulerian oceanic waves. We first use a simple case to get a basic relation between Eulerian wave parameters and Lagrangian ones. Because of the zonal tendency of the studied float trajectories (Fig. 2), we express a simple case of the

current perturbations due to a plane wave propagation with the following system of equations :

$$\begin{cases} u = u_0 \\ v = v_0 \sin\left(\omega_\mathrm{Te} t - k_\mathrm{Te} x + \varphi_\mathrm{Te}\right) \end{cases} \tag{4}$$

where $u_0$ and $v_0$ are zonal and meridional velocities considered as constants, $\omega_\mathrm{Te}$ is the frequency, $k_\mathrm{Te}$ is the wavenumber and $\varphi_\mathrm{Te}$ is the phase shift term of the Eulerian theoretical wave. Note that $v_0$ is the amplitude $A_\mathrm{Te}$ of the Eulerian theoretical wave. By convention, the subscript 'T' is used for theoretical, 'M' for measured, 'e' for the Eulerian description of the wave and '$\ell$'

for the Lagrangian one (Tab. 1). After performing some tests (Appendix A), we choose to set the zonal velocity of the floats as





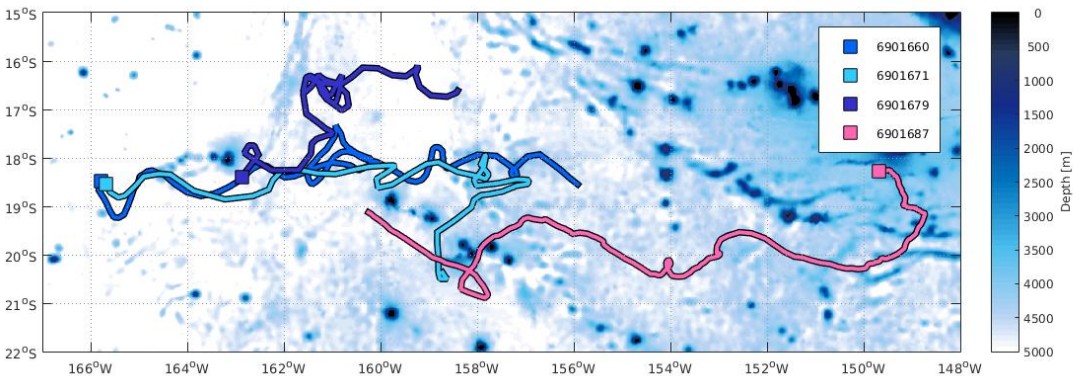

**Figure 2.** Trajectories of the studied floats superimposed on bottom topography (in m). The squares represent the initial location of the floats.

the global mean velocity of the float trajectory:

$$u_0 = \begin{cases} \frac{L^{\text{end}} - L^1}{t^{\text{end}} - t^1} = 1.65 \text{ cm/s} & \text{for float 660} \\ \\ \frac{L^{\text{end}} - L^{58}}{t^{\text{end}} - t^{58}} = -1.93 \text{ cm/s} & \text{for float 687} \end{cases} \quad (5)$$

where $L^{\text{end}}$ is the location of the float at the last cycle. For float 687, the 58th cycle corresponds to the start of the float westward propagation. Thus, we only consider the 687 float trajectory from the 58th cycle to the end location.

5    In such a configuration, the Lagrangian observation of the float can be represented by a Doppler effect on the perception of the theorical wave. So we can express the Lagrangian properties of the theoretical wave ($\omega_{\text{T}\ell}$, $k_{\text{T}\ell}$ and $A_{\text{T}\ell}$) as a function of $\omega_{\text{Te}}$, $k_{\text{Te}}$, $A_{\text{Te}}$ and $u_0$:

$$\begin{cases} \omega_{\text{T}\ell} = \omega_{\text{Te}} - k_{\text{Te}} u_0 \\ k_{\text{T}\ell} = k_{\text{Te}} - \frac{\omega_{\text{Te}}}{u_0} \\ A_{\text{T}\ell} = -\frac{A_{\text{Te}}}{\omega_{\text{Te}} - k_{\text{Te}} u_0} \end{cases} \quad (6)$$

But these equations derive from one common expression, hence this system cannot be solved like a 3 equations - 3 unknowns

10   one. Solving 6 is then not trivial without setting either $\omega_{\text{Te}}$ or $k_{\text{Te}}$. So we need another method to find the Eulerian properties of the theoretical wave that better fits the float trajectories. Hereafter, we use those equations in order to build an index $I$ for the

**Table 1.** Summary of the different notations of the wave properties : frequency $\omega$, wavenumber $k$ and amplitude $A$.

| | |
|---|---|
| $\omega_{\text{M}\ell}$ , $k_{\text{M}\ell}$ , $A_{\text{M}\ell}$ | Measured float wave, Lagrangian description. |
| $\omega_{\text{T}\ell}$ , $k_{\text{T}\ell}$ , $A_{\text{T}\ell}$ | Theoretical wave, Lagrangian description. |
| $\omega_{\text{Te}}$ , $k_{\text{Te}}$ , $A_{\text{Te}}$ | Theoretical wave, Eulerian description. |
| $\omega_{\text{TMe}}$ , $k_{\text{TMe}}$ , $A_{\text{TMe}}$ | Theoretical wave that better fits the measured float waves, Eulerian description. |




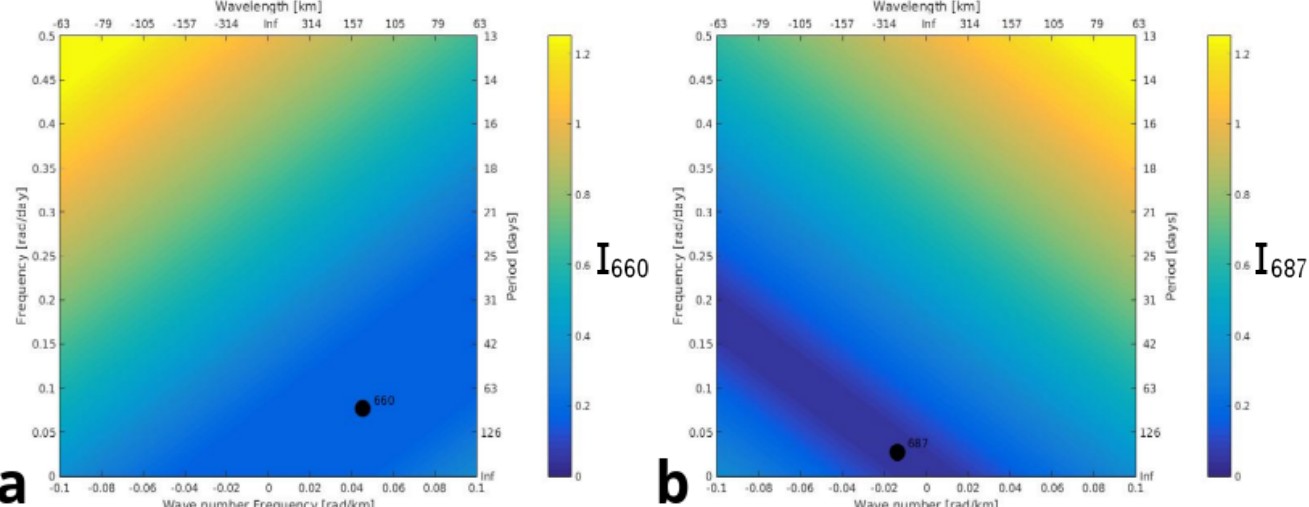

**Figure 3.** Difference between observed Lagrangian float wave parameters and Lagrangian theoretical wave ones (index **I**) for (a) float 660, (b) float 687. Negative wavelengths describe westward waves. Black dots indicate the observed Lagrangian parameters.

differences between the Lagrangian properties of the float waves ($\omega_{\mathrm{M}\ell}$ and $k_{\mathrm{M}\ell}$) and the Lagrangian properties of the potential theoretical waves ($\omega_{\mathrm{T}\ell}$ and $k_{\mathrm{T}\ell}$). $I$ is defined by the following expression:

$$I_a = \left\{ |\omega_{\mathrm{M}\ell} - \omega_{\mathrm{T}\ell}| + |u_0\left(k_{\mathrm{M}\ell} - k_{\mathrm{T}\ell}\right)| \right\}_a \tag{7}$$

where $a$ is the float considered.

We made the calculations for a range of Eulerian frequencies and wavenumbers ($\omega_{\mathrm{Te}}$ and $k_{\mathrm{Te}}$), thanks to Equations 6, in order to get the diagrams of this index for float 660 ($I_{660}$, Fig. 3a) and for float 687 ($I_{687}$, Fig. 3b). For each index, the zone gathering all the index minima has a linear shape : positive for float 660 and negative for float 687. Hence, since these linear slopes are of opposite signs, minimizing the sum of the two indexes leads to find where the two lines of minima cross each over. The crossing approach determines the properties of the theoretical wave that could better fit both the float trajectories.

This method could be used with several other floats and be expressed by the following formulation:

$$I = \sum_{na=1}^{N} I_{na} \tag{8}$$

where $N$ is the total number of floats considered, in our study $N = 2$. Using this numerical approch, we can find all the couples ($\omega_{Te}$, $k_{Te}$) that minimize $I$, and hence that could fit the float trajectories.

    In order to compare the couples that minimize $I$ and find the one that better fits the 660 and 687 float trajectories, we

make a simulation of the idealized trajectory for each couple. The simulations use Equations 4, integrating them to express the



longitude $x(t)$ and the latitude $y(t)$ of Lagrangian particles as a function of time:

$$
\begin{cases}
x(t) = x_0 + u_0 t \\
y(t) = y_0 + \frac{v_0}{\omega_{\text{Te}} - k_{\text{Te}} u_0} \left[ cos\left( \left( \frac{\omega_{\text{Te}}}{u_0} - k_{\text{Te}} \right) x(t) - \frac{w_{\text{Te}}}{u_0} x_0 + \varphi_{\text{Te}} \right) - cos\left( -k_{\text{Te}} x_0 + \varphi_{\text{Te}} \right) \right]
\end{cases}
\tag{9}
$$

where $x_0$ and $y_0$ define the initial position of the float. The simulations run with a time step of 1 day for 642 days ($\left\{ t^{\text{end}} - t^1 \right\}_{660} = \left\{ t^{\text{end}} - t^{58} \right\}_{687} = 642$ days in Equations 5). In order to compare the simulations to the float trajectories, we interpolate the lat-

itudes of the float, $y_{\text{int}}$, at the longitudes of the corresponding simulations. Then we calculate the sum of the differences of latitude using an index $J$, defined as:

$$
J_a = \left\{ \sum_{j=1}^{m} |\, y_{\text{int}}(j) - y_{\text{simu}}(j) \,| \right\}_a
\tag{10}
$$

where $a$ is the considered float, $y_{\text{simu}}$ are the latitudes of the simulation and $m$ is the total number of days, here $m = 642$. Because the value of $\varphi_{\text{Te}}$ can influence the value of $J$, we made several simulations for each couple ($\omega_{\text{Te}}$ , $k_{\text{Te}}$) varying $\varphi_{\text{Te}}$

from $-\pi$ to $\pi$ with a step of $\frac{1}{10}\pi$. As for index $I$, we can sum $J_{660}$ and $J_{687}$ to get the couple that minimizes the trajectory differences of both floats 660 and 687. This method could also be used with several other floats and be expressed by the following equation:

$$
J = \sum_{na=1}^{N} J_{na}
\tag{11}
$$

These two steps give us the couple ($\omega_{TMe}$ , $k_{TMe}$) of the theoretical wave that better fits the trajectories of floats 660 and

687. This couple can now be compared to classical Eulerian oceanic waves such as Kelvin, Rossby and Kelvin-Helmholtz instability waves.

## 2.3 Ancillary data

In order to evaluate the possible link between oxygen anomalies and surface currents in the Coral Sea, we use the AVISO altimetry products (www.aviso.altimetry.fr). We select ocean-level elevation anomaly products, reworked from all satellites,

to obtain geostrophic velocity anomaly fields ($u_g$ and $v_g$) with a 1/4 degree resolution. From those data we calculate the geostrophic velocity amplitude. We also use HYCOM (HYbrid Coordinate Ocean Model) re-analysis GLBu0.08 of the experience 91.1 from March 2015 to March 2016 (www.hycom.org). This system is a hybrid isopycnal-sigma-pressure (generalized) coordinate ocean model with 1/12 degree horizontal resolution and 40 vertical levels, assimilating in situ data.

In order to replace our analyses within the global thermohaline context, we used the ISAS13 atlas (In Situ Analysis System,

Gaillard, 2012) that provides a climatology of thermohaline properties. This atlas collects and processes all the profiles provided by the Argo floats from 2004 to 2012 in order to calculate a monthly global climatology over the entire depth of the oceans. More details are provided by Gaillard et al. (2016). We apply the same methods to calculate the density as those used for HYCOM re-analysis.



In the following, whatever is the source for the temperature and salinity fields, the density of the water masses is computed with the Matlab toolbox Gibbs-Seawater based on the TEOS-10 convention (www.teos-10.org/software.htm).

## 3 Results

### 3.1 Coral sea

The trajectory of float 656 can be clearly associated with the flow along the New Caledonian coasts at the entrance of the Coral Sea (Gasparin et al., 2011, Fig. 1a). The float then travels westward until the Queensland plateau where it often reaches the bottom before stabilizing near its parking depth. Hence, from November 2015 to October 2016, it either moves very slowly or stays stuck in the same region. Afterwards the float moves northwestward circumventing the plateau at the end of the period analyzed here. This trajectory corresponds very strongly with the NCJ pathway. Here we focus on the measurements made by
this float.

#### 3.1.1 The DOXY variability of the jets

In addition to the standard parameters of conservative temperature and absolute salinity, float 656 provides DOXY profiles. We plot the temporal evolutions of these properties, corresponding to a mainly East to West transect (Fig. 4). Whereas temperature and salinity vary quite homogeneously below the upper layers, we can observe a strong oxygen stratification of the water
column: higher concentrations at the surface (around 200 $\mu$mol.kg$^{-1}$) and between 500 m and 900 m (around 190 $\mu$mol.kg$^{-1}$) and lower concentrations between 100 m and 500 m, as well as below 900 m (around 170 $\mu$mol.kg$^{-1}$). Stratification is only partially found in the salinity profiles: lower concentrations at the surface (around 35.5 g.kg$^{-1}$) and below 400 m (less than 35.25 g.kg$^{-1}$) and higher concentrations between 100 m and 400 m (around 36 g.kg$^{-1}$).

In the intermediate layers, between 100 and 500 m, we notice a strong variability in the oxygen, and especially strong pulses
of low values. We isolated two strong deoxygenation events (D1 and D2) with values below 140 $\mu$ mol.kg$^{-1}$. D1 is an event of about one month covering October 2015 with measurements between 150 $\mu$mol.kg$^{-1}$ and 140 $\mu$mol.kg$^{-1}$. D2 is spread over two months from February to March 2016 with measurements lower than 130 $\mu$mol.kg$^{-1}$. The DOXY signatures of D1 and D2 do not correspond to the classical characteristics of the NCJ (Gasparin et al., 2011), whereas the float is exactly on its pathway. We therefore seek to know its origin. Based on the work of Gasparin et al. (2014) and Rousselet et al. (2016), we
hypothesize that this signature originates from the NVJ waters whose theoretical and observed pathway is located 4$^o$ further north. The figure of Appendix B shows the depth variability of the DOXY minimum of each profile during D2 and helps to understand the link between the deoxygenation events and the properties of NVJ waters.

#### 3.1.2 Thermohaline and circulation context

We focus on D2 that is longer and stronger than D1. The geostrophic velocity fields from AVISO (Fig. 5a) allow us to visualize
the position of the float during D2 in relation to the surface circulation. We notice that, during the whole duration of D2, the



**Figure 4.** Profiles of float 656 over depth and time for (a) absolute salinity, (b) conservative temperature and (c) dissolved oxygen concentration. Every colored point corresponds to a measurement. The horizontal black lines indicate the isopycnals from 1024 to 1031 kg.m$^{-3}$. The vertical black lines indicate the limits of the deoxygenation events D1 and D2. Over the top of the pictures, $\Delta t$ precises the two different sampling frequencies.





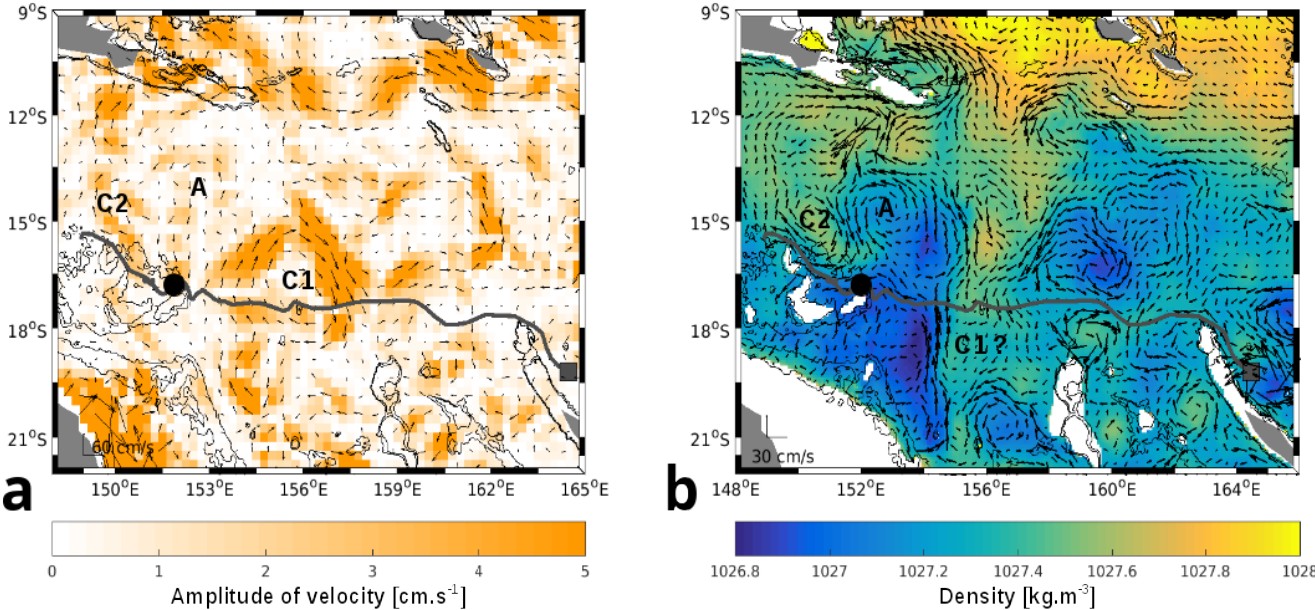

**Figure 5.** (a) Amplitude of surface geostrophic velocity on February 29th during D2, 2016 calculated from AVISO product ; (b) Density and velocity field at 300 m depth on February 29th during D2, 2016 calculated from HYCOM re-analysis. The gray line corresponds to the full trajectory of float 656 and the black point to the location of the float at the stated date. The black lines correspond to the bottom topography with 300 m and 1000 m depth.

float is located on the Queensland plateau where we can note that the deepest measurements are shallower than the parking depth (also visible in Figures 4a,b,c). This leads to short displacements over the whole event, making the interpretation of the results easier. We can identify a very large cyclonic structure to the east of the plateau centered at $156^o$E, that we name C1, and a much weaker cyclonic structure, C2, located to the north-west of the plateau. Between them, we observe an anticyclonic

5     structure, A, centered at $153^o$E and stretching from $13^o$S to $17^o$S. The western branch of A ($152^o$E) forms a meridional path with southward velocities.

     Using the HYCOM re-analysis, we can compare the AVISO observations to the modeled velocity field at different depths. At 300 m, the depth of the DOXY minimum, we can clearly identify structures C2 and A but C1 is harder to locate (Fig.5b). C1 may be located $1^o$ farther south. A is much more circular in the HYCOM re-analysis than in the AVISO surface data. These

10     data also allow us to consider the densities of these water masses and thus enable us to track the potential NVJ waters down to the NCJ pathway. Figure 5b clearly shows that the NVJ waters can be associated with C2 for instance.

     Otherwise, studying the AVISO product and HYCOM re-analysis for D1, we do not identify an eddy structure or a circulation shape that could explain that DOXY anomaly. Nevertheless, due to the large scale features of water masses and the low values of DOXY, observed in Figure 4c, we could hypothesized that such deoxygenation events are related either to the intrusion of





waters transported by mesoscale eddies or by NVJ meanders. Because we can observe such circulations with AVISO products, they definitely affect the surface layers and then impact the studied diazotrophic zones of the OUTPACE cruise. In the section 4.1, we will further discuss such aspects.

It should be noticed that it still remains difficult to interpret observations of different nature, *i.e.* from an Eulerian versus Lagrangian point of view, and that further work is required to replace the float observations in their complete dynamical context.

## 3.2 Central Pacific Ocean

### 3.2.1 Float waves description and Doppler shift correction

In this part we focus on the trajectories of three OUTPACE floats : 660, 671 and 679 moving eastward and one THOT float : 687 moving westward. These floats were all deployed in spring 2015 and now show, after two years of drifting, zonal wavy trajectories (Fig. 2). We consider this group of OUTPACE floats distinctly from the THOT float because of their different directions. Then, we compare their respective float wave characteristics.

The wave characteristics of the two groups (Tab. 2) are clearly different: 71 days period and 159 km wavelength for the eastward group and 232 days period and 434 km wavelength for the westward float. Based on the couple ($\omega_{M\ell}$ , $k_{M\ell}$) and following the method described in section 2.2, we first estimate the Eulerian characteristics of the theorical waves ($\omega_{Te}$, $k_{Te}$ and ultimately $\omega_{TMe}$, $k_{TMe}$) that better fit the floats.

To simplify the presentation, we only use the 660 and 687 float waves properties. We choose these two floats because they are the ones with the most regular trajectories (neither going northward like float 679 or southward like float 671). As explained in section 2.2, since the two regions of minima cross, the two observed float waves can be the signature of a single theoretical wave. Considering this hypothesis, we find that this wave is defined in the ranges from 0 to 0.075 rad.day$^{-1}$ for the frequency and from -0.04 to 0.02 rad.km$^{-1}$ for the wavenumber (Fig. 6a). In order to get a better resolution and minimize the calculation time, we make a zoom on those ranges of frequencies and wavenumbers before doing the minimum calculation. We set a resolution of $5 \cdot 10^{-5}$ rad.day$^{-1}$ and $5 \cdot 10^{-5}$ rad.km$^{-1}$ and find 8083 minima, mostly in the westward region (Fig. 6b).

In order to compare the couples ($\omega_{Te}$ , $k_{Te}$) to classical oceanic waves, we calculate the dispersion equation of Kelvin and Rossby in a vertical barotropic case, Kelvin-Helmholtz instability wave in a two layers case and Rossby waves in several baroclinic cases (the different cases are explained in Appendix C). We observe that the characteristics of the Kelvin and Kelvin-Helmholtz instability waves are not in the same ranges as the ones of the couples we want to identify. Rossby waves (**R**) are the ones that better fit them. The barotropic case **R$_b$** is out of range but most solutions are located around the curves of the baroclinic cases with a thermocline at 35 m (**R$_{35}$**) and 200 m (**R$_{200}$**).

Using the $J$ index, we are able to select the couple that better fits both the 660 and the 687 float trajectories. The results give us a wave of 160 days period and 855 km westward wavelength. Figure 7 illustrates the agreement between the observed and the reconstructed trajectories obtained from it. Obviously, some other processes with small-scale signatures also influence the observed float trajectories. Nevertheless, a single wave, added to the float respective zonal background currents, can mainly





**Table 2.** Period $T$ and wavelength $\lambda$ of the float waves. The mean, standard deviation and extrema values are calculated for the eastward floats : 660, 671 and 679 (top of the table). Then details are given for each floats: dates, geographical area, half periods, half wavelengths and corresponding mean and standard deviation values for $T$ and $\lambda$.

| | EASTWARD (stat) | | | |
|---|---|---|---|---|
| | Lat [$^o$S] | Lon [$^o$W] | $T$ [days] | $\lambda$ [km] |
| Mean $\pm$ std | $18.8 \pm 0.9$ | $160.8 \pm 3.4$ | $71 \pm 21$ | $159 \pm 74$ |
| [min;max] | [20.7 ; 17.8] | [168.2 ; 156.7] | [40 ; 120] | [42 ; 275] |

| | | | EASTWARD (details) | | | | |
|---|---|---|---|---|---|---|---|
| Float | Date | Lat [$^o$S] | Lon [$^o$W] | $T/2$ [days] | $\lambda/2$ [$^o$] | $\bar{T}$ [days] | $\bar{\lambda}$ [km] |
| | | | | 35 | 0.7 | | |
| | | | | 42 | 1.4 | | |
| | Feb 7th, 2016 | | | 40 | 0.5 | | |
| 660 | to | [18.6 ; 17.8] | [161.4 ; 156.7] | 30 | 0.2 | $82 \pm 20$ | $138 \pm 74$ |
| | Nov 21st, 2016 | | | 35 | 0.8 | | |
| | | | | 45 | 0.6 | | |
| | | | | 60 | 0.5 | | |
| | | | | 30 | 0.6 | | |
| | | | | 30 | 1.1 | | |
| | Jul 8th, 2015 | | | 40 | 1.2 | | |
| 671 | to | [18.9 ; 18.1] | [162.6 ; 157.8] | 40 | 0.4 | $70 \pm 11$ | $201 \pm 74$ |
| | Feb 3rd, 2016 | | | 40 | 1.3 | | |
| | | | | 30 | 1.0 | | |
| | | | | 20 | 0.7 | | |
| | Aug 13th, 2016 | | | 20 | 0.6 | | |
| 679 | to | [19.2 ; 17.3] | [161.5 ; 158.8] | 30 | 0.7 | $56 \pm 26$ | $127 \pm 32$ |
| | Dec 31st, 2016 | | | 20 | 0.3 | | |
| | | | | 50 | 0.6 | | |
| | | | WESTWARD (details) | | | | |
| | Jul 18th, 2015 | | | 65 | 2.5 | | |
| 687 | to | [20.9 ; 19.2] | [157.8 ; 150.1] | 95 | 1.2 | $232 \pm 93$ | $434 \pm 159$ |
| (THOT) | Oct 31st, 2016 | | | 173 | 2.8 | | |
| | | | | 130 | 1.5 | | |



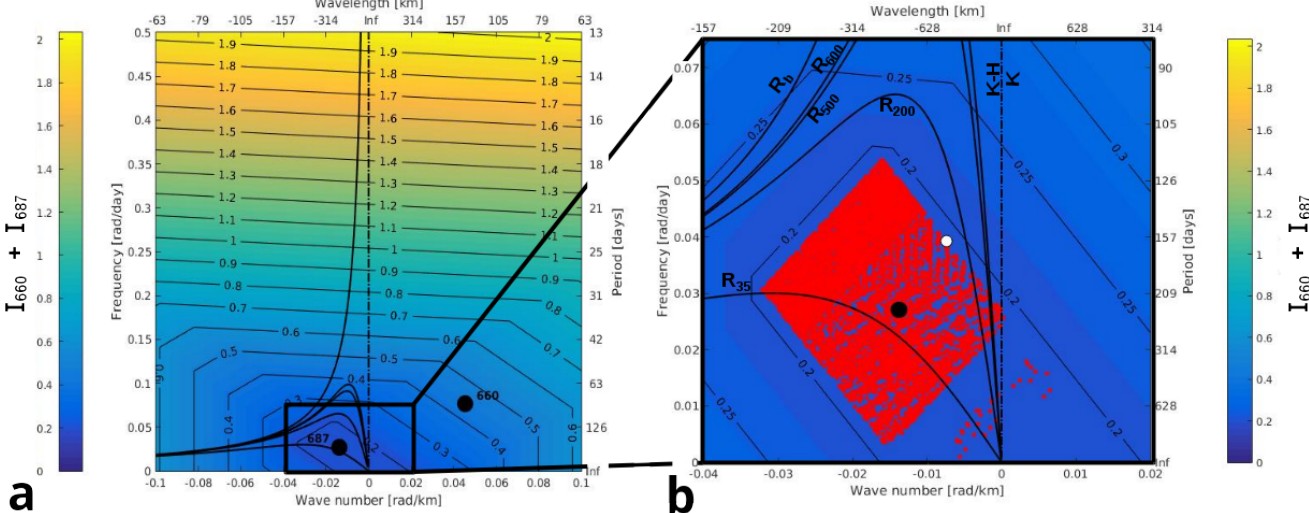

**Figure 6.** Difference between observed Lagrangian float wave characteristics and Lagrangian theoretical wave ones (index **I**) for (a) the sum of floats 660 and 687 and (b) the associated minimum calculations with regard to dispersion equations of Rossby wave (solid line), Kelvin wave (dotted line), Kevin-Helmholtz instability wave (break line). Subscript *b* means a barotropic case, *200* means a baroclinic case with a stratification at 200 m. The negative wavelength describes a westward wave. Black dots indicate the observed Lagrangian parameters, on (b) only 687 ones are inside the range; the red dots show the minima of the index hence the best parameters that can explain both trajectories; the white dot is the couple ($\omega_{TMe}$, $k_{TMe}$) that better fits the 660 and 687 float trajectories.

explain the two float trajectories that would otherwise be classified, at first sight, as behaving differently. Such observations of a potential plane wave have been rarely highlighted so far, and even less at such depth.

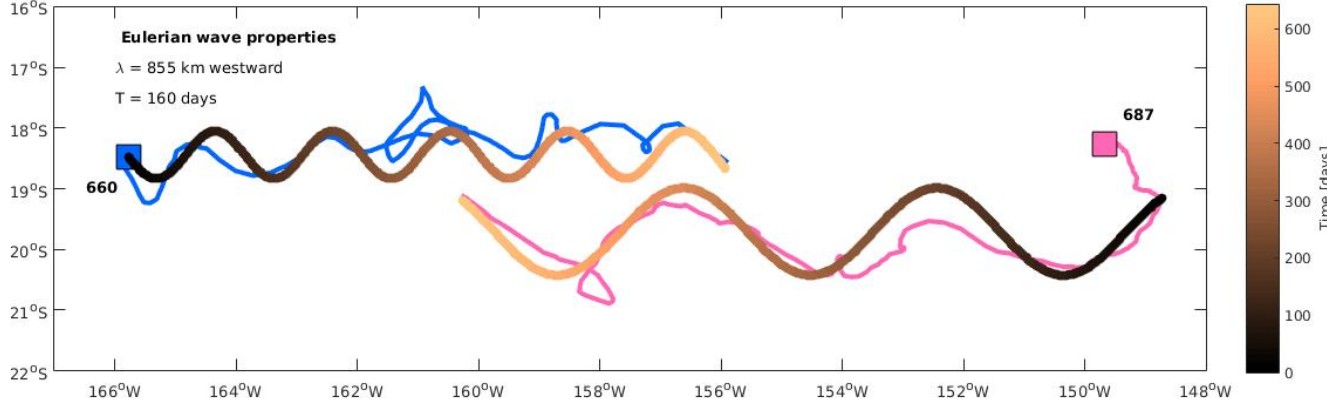

**Figure 7.** Best analytical solution of Lagrangian trajectories for both floats 660 and 687. The blue line refers to float 660, the pink one to float 687, the squares refer to their inital location. The black to brown lines refers to the best theoretical solution with a time step of one day.





**Table 3.** Meridional slope of the isopycnal 1031.95 kg.m$^{-3}$ in the monthly climatology of the ISAS13 atlas.

| Month | Jan. | Feb. | Mar. | Apr. | May | Jun. | Jul. | Aug. | Sep. | Oct. | Nov. | Dec. |
|---|---|---|---|---|---|---|---|---|---|---|---|---|
| Slope [m.$^o$lat$^{-1}$] | 4.3 | 4.5 | 4.2 | 4.0 | 4.3 | 3.9 | 4.3 | 4.0 | 4.0 | 4.4 | 4.1 | 4.4 |

### 3.2.2 Thermohaline and circulation context

Using HYCOM re-analyses, we are able to replace the trajectory of the floats into the circulation at 1000 m depth. Figures 8 a and b highlight the current striations as mentioned in the introduction. We observe that the modeled striations have, on average, widths of 1$^o$ to 1.5$^o$ of latitude, which are smaller than those observed by Ollitrault and Colin de Verdière (2014) (Fig. 1c). The

mean shear of zonal velocity observed by the floats 660 and 687 is equal to 4.2 cm.s$^{-1}$ (zonal velocities used in the Appendix A for case b). We note a difference between the Eulerian modeled zonal velocities and the Lagrangian zonal velocities of the floats, which is not surprising taking into account the Eulerian versus Lagrangian description.

We also use the HYCOM re-analysis fields of temperature and salinity in order to calculate the density context of the studied area (Fig. 8c). The two floats 660 and 687 are both at the interface of two different water masses : the northern one with

a density around 1032 kg.m$^{-3}$ and the southern one with a density around 1031.85 kg.m$^{-3}$. This corresponds to a density difference of 0.15 kg.m$^{-3}$ in a 2$^o$ latitudinal range with the isopycnals being deeper southward. The front location, shape and intensity are variable with some extreme events of displacement or intensity, like on November 13th, 2015 (Fig. 8d). But the front globally stays a zonal boundary located around 20$^o$S, like on September 29th, 2015 (Fig. 8c).

Meridional cross-sections of HYCOM re-analysis (Fig. 9a) show a salinity gradient around 900 m: saltier (35 g.kg$^{-1}$) north

of 19$^o$ and less salty (34.5 g.kg$^{-1}$) south of 19$^o$. These values are also observed in the ISAS13 climatology (Fig. 9b). To quantify the meridional slope of this front, we analyze the depth of the isopycnal 1031.95 kg.m$^{-3}$ (average between the two sides of the front) from 147$^o$W to 167$^o$W and from 15$^o$S to 25$^o$S. Then we perform a linear regression on the isopycnal depths in their steepest part between 17$^o$ S and 22$^o$ S. We can obtain, for example, low values: 3.8 m per degree of latitude on November 13th, 2015 (Fig. 8d), average values: 4.2 m per degree of latitude on September 29th, 2015 (Fig. 8c), and what

we call extreme values such as 26.9 m per degree of latitude on November 13th, 2015. To verify if the meridional slope varies over the year, we have done the calculations for the complete ISAS13 climatology (Tab. 3). These slope values range from a minimum of 3.9 m per degree of latitude in June to a maximum of 4.5 m per degree of latitude in February. They do not exhibit any apparent annual cycle and the density front appears to be a permanent feature.

Otherwise, we note that, from May to July 2015, a cyclonic eddy (centered about 165$^o$W and 18.5$^o$S on Figures 8a,b)

crossed the trajectory of float 660. Hence, it can not be ruled out that this eddy could have influenced the trajectory of float 660 and produced a certain variability in its oscillations. Concerning the trajectory of float 687, we do not observe any stable eddy structure around it during all its studied period. That explains why we did not choose to interpret the wavy float trajectories with the unique hypothesis of eddy impacts.





**Figure 8.** (a,b) Zonal velocity (positive for eastward velocity) at 1000 m depth on (a) July 1st, 2015 and (b) July 16th, 2015 from HYCOM re-analysis; (c,d) density and velocity field at 1000 m depth on (c) September 29th, 2015 and (d) November 13th, 2015 calculated from HYCOM re-analysis; The gray lines correspond to the full trajectory of the floats and the black portions to the location of each float at the stated date and on the 15 following days.




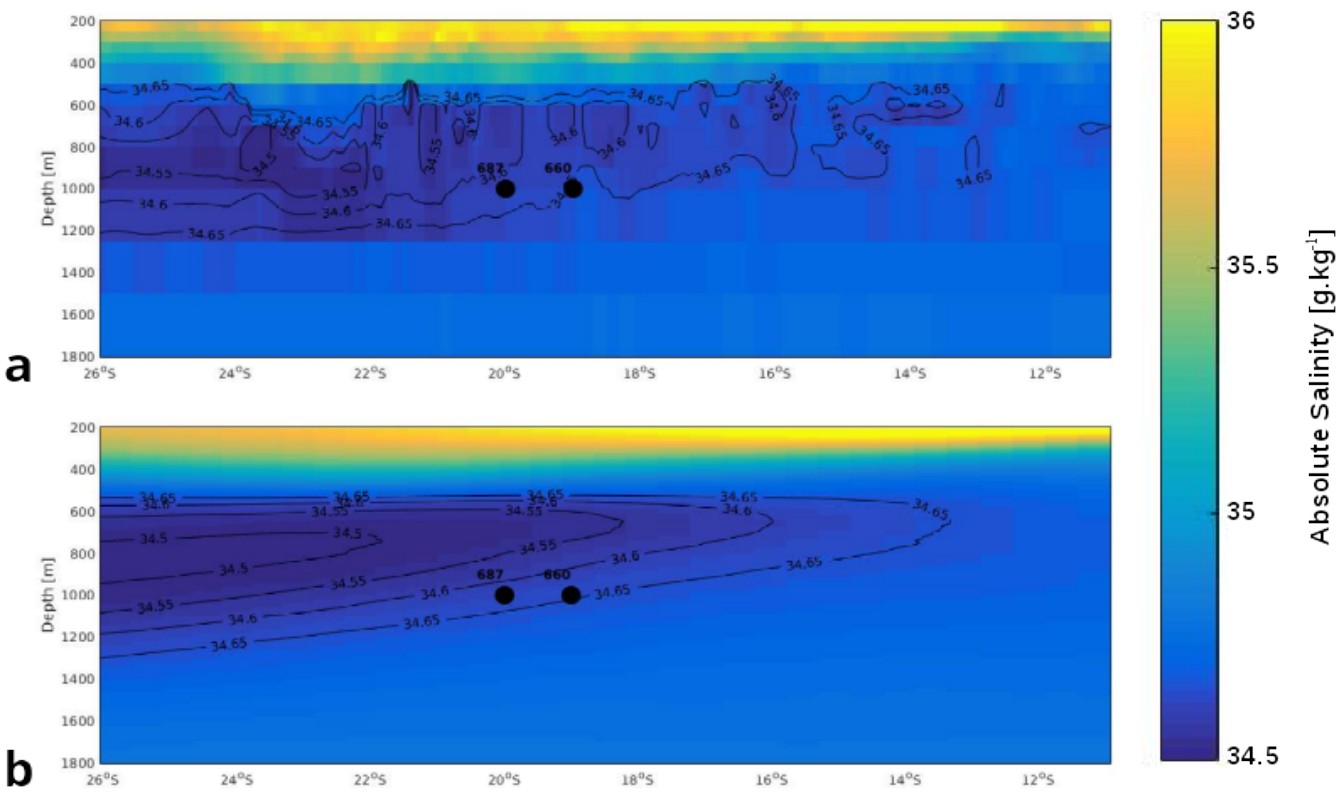

**Figure 9.** Absolute salinity meridional section at $157^{\circ}W$ calculated from (a) HYCOM re-analysis on November 13th, 2015 and (b) ISAS13 atlas for November seasonal climatology. Black dots indicate the mean location of the floats.

## 4 Discussion

### 4.1 Coral Sea

Analyzing the DOXY measurements of float 656, we were able to describe two distinct events with well marked oxygen minima (D1 and D2, Fig. 4) between 150 m and 400 m depth, *i.e.* near the upper part of the main thermocline. Inspired by the previous study of Rousselet et al. (2016), we associate these oxygen minima to the NVJ waters, much more deoxygenated than those of the NCJ. We propose that the NVJ waters were indeed transported by mesoscale eddies. To support this hypothesis, we compare the geostrophic currents at the surface with AVISO data and also use the HYCOM reanalyses at 300 m. Even if the comparison was not clear enough for D1 to be fully conclusive, in the case of D2 we have been able to identify a cyclonic structure (C2, Fig. 5) and an anticyclonic structure (A1) in the two sets of data. These vortex structures are involved in the transport of NVJ waters southward to the NCJ pathway. Maps of density resulting from the HYCOM re-analysis inform us about the nature of the cores of these vortex structures. Structure A carries NCJ waters while C2 transports NVJ waters. This



clear difference forms a density gradient of 0.6 kg.m$^{-3}$ over 3$^o$ of longitude approximately. From this, we can hypothesize that the signature of the oxygen minimum is due to C2 carrying NVJ waters in its core. Otherwise, the common branch of C2 and A forms a local southward current exactly toward the position of the float 656. Hence, we can also make a second hypothesis that the NVJ waters could be carried by the northern branch of A and then be transported southward thanks to the current located

between A to the east and C2 to the west. The first hypothesis fits the results of (Rousselet et al., 2016), with the difference that the structure transporting NVJ waters is cyclonic and not anticyclonic. Hence the two possibilities that we propose widen the comprehension of the connection processes between the NVJ and the NCJ and claim for the explicit consideration of mesoscale eddies variability in future modeling approach.

    Thanks to the analysis of the OUTPACE observations, Fumenia et al. (2018) hypothesize that the location of nitrogen sources

and sinks could be decoupled. Thus, the authors propose that the transport of rich nitrogen thermocline waters from N$_2$ fixation could join the subtropical gyre through the EAC (Fig. 1a). Bouruet-Aubertot et al. (2018) also observe a westward increase of turbulences during OUTPACE. This leads to a strong turbulent regime in the Melanesian Archipelago, located at the entrance of the Coral Sea, highly visible in nitrogen measurements during the long term stations. Extending these conclusions to the Coral Sea set a favorable context to the exchanges between NVJ and NCJ. Specific exchanges from NVJ to NCJ could, then,

strengthen the recirculation in the subtropical gyre of rich nitrogen waters. Thus understanding their dynamics could help us to better understand their impact on the propagation of biogeochemical components.

## 4.2   Central Pacific Ocean

The trajectories of OUTPACE and THOT Argo floats give us two groups of float waves with different characteristics. Whereas the long-term mean displacement of the floats could be explained by the presence of alternating striations as deduced from the

displacement of the Argo floats at their parking depth, we focused on their quite-surprising oscillation characteristics. In this study, we show that their oscillating trajectories can be caused by a single theoretical wave of 160 days period and 855 km wavelength superimposed to the zonal background current. Focusing on Figure 7, the fit over the trajectory of float 687 is very convincing. It is less so for float 660. We can explain this by recalling that the beginning of the 660 trajectory is, partially or entirely, influenced by an eddy passing through (Fig. 8a,b). Another explanation could be that, in Eq. 4 and 5, we consider the

zonal velocity $u_0$ as constant during the entire simulations. But a variable zonal velocity is likely to strongly impact the float trajectory. We detail this point with two examples that can be observed on the 660 and 687 trajectories, respectively. The first one takes place on the 660 trajectory between 159$^o$W and 158$^o$W (Fig. 7). It happens when the zonal velocity decreases while keeping the same direction eastward (Fig. 10a). This leads to a smaller local wavelength. The second one takes place on the 687 trajectory around 158$^o$W. It happens when the zonal velocity changes direction, from westward to eastward (Fig. 10b).

This leads to a loop in the Lagrangian trajectory. The impact of a variable zonal velocity does not refute our hypothesis of a single wave influencing the two floats 660 and 687, but questions the precision of the values found ($\omega_{TMe}$, $k_{TMe}$). The same comment also applies to the description of the couples $(\omega_{M\ell}, k_{M\ell})_{660}$ and $(\omega_{M\ell}, k_{M\ell})_{687}$ that could not be performed using a Fourier transformation or a wavelet analysis due to the short duration of the float data time series compared to the sampling



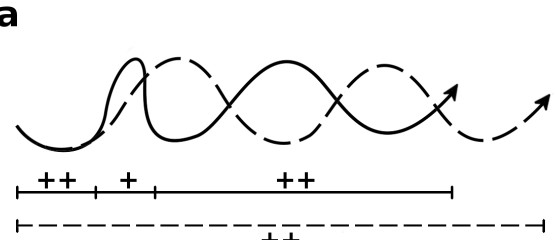

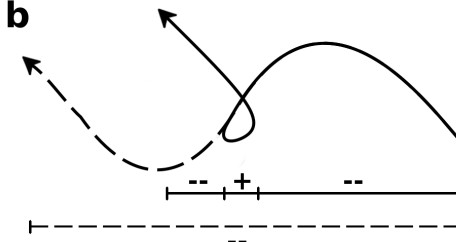

**Figure 10.** Sketch explaining the influence over a Lagrangian trajectory of (a) a zonal velocity decrease like in the 660 trajectory and (b) a zonal velocity inversion like in the 687 trajectory. The horizontal dash lines refer to the trajectories influenced by a wave with a constant zonal velocity. The horizontal solid lines are segmented with the periods associated to the sign above them. The signs indicate the strength and the direction of the zonal velocity : ++ for strong eastward, + for weak eastward and - - for strong westward.

frequency. Using a plane wave description is a source of inaccuracy because such wave are rarely observed in open ocean studies. Some propositions for further analyses are detailed in the next section.

Based on the HYCOM re-analysis, we confirm that both the 660 and the 687 floats move in shear environments. This shear context is supported by a density front which is permanent all year round. This front could be associated with the base of
the northern edge of the Antarctic Intermediate Waters that reach this latitude near our study zone (Bostock et al., 2013). The best fitting wave that we found is close to a Rossby wave in a baroclinic ocean case with a thermocline at 200 m. This case is close to the real context of the studied zone where the upper thermocline waters are located around 200 m (GLODAPv2 database, Fumenia et al., 2018). Moreover, this zone is precisely located where Rossby waves are frequently found and their signatures can be observed through sea level anomalies (Maharaj et al., 2007). Those arguments strengthen the hypothesis of
a Rossby wave close to 160 days period and 855 km wavelength. These observations lead to a large discussion of the possible interactions between waves and intermediate flows.

Such processes could have a significant impact on the sampling strategy of intermediate waters by shifting the front location hundreds of kilometers from its mean location. OUTPACE sampling strategy during long duration stations keep sampling within Rossby radii of each station ensuring to generally follow the same water mass (de Verneil et al., 2017a). It also questions
the use of averaged datasets of intermediate flows that do not account for such processes. More generally, this study considers the trajectories of floats worth being studied by themselves and beckons the development of different methodologies to exploit such measurements in the future.

## 5  Conclusions & Perspectives

In this study, we have been able to highlight different pattern of the intermediate flows in the South Tropical Pacific Ocean
through the description of the characteristic trajectories of some autonomous floats of the international Argo project deployed during the OUTPACE cruise (Moutin et al., 2017) and by the THOT project (Martinez et al., 2015).





Relying on the measurements of dissolved oxygen concentration (DOXY) onboard a PROVBIO float, we have found a meridional southward transport of NVJ waters by mesoscale eddies causing oxygen minima intrusions in the NCJ pathway around 300 m depth. Our results widen those of Rousselet et al. (2016), since we show that the transport can occur inside the core of a cyclonic eddy or between a couple of anticyclonic and cyclonic structures. Moreover, the hypothesis that the waters

can be transported on the edge of several eddies is a new proposed pathway, as far as we know. To ensure the mechanisms of this meridional transport, the fronts between the described cyclonic and anticyclonic eddies could be studied using Finite Size Lyapunov Exponents calculations. These calculations could be applied to both AVISO and HYCOM data to provide information at different depths. Another solution is to replace the Lagrangian observations in their finer circulation context; a study based on dynamic attractors (Mendoza and Mancho, 2010) could be useful to test our assumptions. Otherwise, since

DOXY is the parameter that can differentiate NVJ from NCJ, recourse to biogeochemical models could help visualizing and studying the connection between the NVJ and the NCJ. Finally, how these differences in the partition between the jets influence the other biogeochemical parameters and the phytoplankton biology remain to be specifically investigated.

In addition to the description of water masses properties, we observe dynamical features such as wavy float trajectories in the central part of the Pacific Ocean corresponding to a circulation mechanism occurring near 1000 m depth (parking

depth). The shear of zonal velocity ($\Delta u = 4.2$ cm.s$^{-1}$ for the floats) associated with a permanent density front ($\Delta \rho = 0.15$ kg.m$^{-3}$) form a favorable context for the development of instabilities. Correcting the impact of the Lagrangian observation in a simplified case, we concentrated on two floats, heading in opposite zonal directions. We found that their behaviors, apparently opposite, could be described with a single wave of 160 days period and 855 km wavelength heading westward. This couple of parameters can be identified as a Rossby wave in a baroclinic context for a two layers ocean with a thermocline at 200

m. To get a better float wave description, using Fourier transformation or a wavelet analysis, would require to wait until the time series are longer. In addition, to refine the method and results, we suggest to improve the Lagrangian simulation, that we used to fit the trajectories, by taking into account the temporal variations of zonal velocity for each float. Whereas, to improve the comparison to classical waves, the dispersion equation of a Rossby wave could be calculated in the 3D context of the interface between Antarctic Intermediate Waters and the North Pacific Deep Waters. The baroclinic instability of the density

front is an alternative hypothesis that could also be considered in order to explain the float trajectories. Because the front is permanent, other immersions of floats at different latitudes on two meridians enclosing the study zone would also consolidate the observations of this process. Nevertheless, all these perspectives are beyond the scope of this paper and will be considered in future works.

Thanks to this study, we underline the importance of eddies as well as waves in the mesoscale dynamics of intermediate

flows. We stress the importance of taking into account individual float measurements and trajectories in further studies in order to better understand water mass transport, mixing processes and their potential impacts on biogeochemical cycles.





## Appendix A: Wave fit on float trajectories

**Table A1.** Characteristics of the theoretical wave that better fit both 660 and 687 float trajectories for different zonal speed cases and resolution. Case a) refers to the global mean zonal velocity of each float from the first cycle to the last one for 660 trajectory and from the 58th cycle to the last one for 687 trajectory (see section 2.2). Case b) refers to the mean of all the measured zonal velocities for each float. Case c) refers to the mean of all floats (*i.e.* the mean of all the values obtained in case b).

| Case | Zonal velocity 660 | 687 | Identical resolution for $\Delta\omega$ and $\Delta k$ | Potential fits | $T_{TMe}$ | $\lambda_{TMe}$ | $\varphi_{TMe}$ | Propagation |
|------|------|------|------|------|------|------|------|------|
| | [cm/s] | | [rad/day & rad/km] | [#] | [days] | [km] | [rad] | |
| | 1.65 | -1.93 | $5 \cdot 10^{-4}$ | 107 | 126 | 449 | $0.7\pi$ | Westward |
| | 1.65 | -1.93 | $2 \cdot 10^{-4}$ | 502 | 126 | 449 | $0.7\pi$ | Westward |
| a) | 1.65 | -1.93 | $1 \cdot 10^{-4}$ | 2054 | 121 | 422 | $0.7\pi$ | Westward |
| | 1.65 | -1.93 | $7 \cdot 10^{-5}$ | 4592 | 121 | 417 | $-0.5\pi$ | Westward |
| | 1.65 | -1.93 | $5 \cdot 10^{-5}$ | 8083 | 160 | 855 | $-0.5\pi$ | Westward |
| | 1.83 | -2.34 | $5 \cdot 10^{-4}$ | 121 | 182 | 393 | $0.1\pi$ | Westward |
| | 1.83 | -2.34 | $2 \cdot 10^{-4}$ | 715 | 150 | 308 | $-0.5\pi$ | Westward |
| b) | 1.83 | -2.34 | $1 \cdot 10^{-4}$ | 2945 | 169 | 332 | $0.1\pi$ | Westward |
| | 1.83 | -2.34 | $7 \cdot 10^{-5}$ | 5801 | 147 | 306 | $0.1\pi$ | Westward |
| | 1.83 | -2.34 | $5 \cdot 10^{-5}$ | 11593 | 169 | 332 | $0.1\pi$ | Westward |
| | 2.09 | 2.09 | $5 \cdot 10^{-4}$ | 2731 | 483 | 838 | $0.3\pi$ | Eastward |
| | 2.09 | -2.09 | $2 \cdot 10^{-4}$ | 16903 | 204 | 383 | $-0.1\pi$ | Westward |
| c) | 2.09 | -2.09 | $1 \cdot 10^{-4}$ | 67591 | - | - | - | - |
| | 2.09 | -2.09 | $7 \cdot 10^{-5}$ | 137567 | - | - | - | - |
| | 2.09 | -2.09 | $5 \cdot 10^{-5}$ | 270730 | - | - | - | - |





## Appendix B: Profiles of float 656 during D2 deoxygenation event

**Figure B1.** DOXY versus absolute salinity profiles of float 656 during D2 (colored lines) compared to characteristic NCJ and NVJ profiles made during BIFURCATION cruise (scaled dots). The grey dots are the other measurements made by float 656. All the profiles are shown from the isopycne 1026 kg.m$^{-3}$ to the one 1030 kg.m$^{-3}$ (*i.e.* between 200 and 650 m depth, approximately).





## Appendix C: Linear Rossby wave dispersion equation

We describe the Rossby wave in different cases. First, in a barotropic ocean, so the dispersion equation is:

$$\omega = -\beta \frac{k}{k_H^2} \tag{C1}$$

with $\omega$ the frequency, $k$ the wavenumber and $k_H$ the horizontal wavenumber of the wave and $\beta$ the Rossby parameter. We place ourselves at 19$^o$S, so $\beta = 2.1 \cdot 10^{-11}$ m$^{-1}$.s$^{-1}$ . In our study we are not able to estimate the value of the meridional wavenumber, hence we neglect it and consider that $k_H = k$ (Maharaj et al., 2007 also use this hypothesis in their study for calculation purpose). The equation becomes:

$$\omega = \frac{\beta}{k} \tag{C2}$$

Second, we take a baroclinic ocean with two layers separated by the thermocline. Still considering $k_H = k$, the dispersion equation is:

$$\omega = \frac{-\beta k}{k^2 + \frac{f_0^2}{c^2}} \tag{C3}$$

with $c$ the phase speed, $f_0$ the Coriolis parameter at 19$^o$S equal to $-4.7 \cdot 10^{-5}$ s$^{-1}$ . In this case the phase speed of the wave is expressed by the following equation:

$$c^2 = \frac{g \Delta \rho \, h_1 h_2}{\rho_2 h_1 + \rho_1 h_2} \tag{C4}$$

with $\rho_1$ and $\rho_2$ respectively the densities of the upper and lower ocean layer, $\Delta \rho = \rho_2 - \rho_1$ and $h_1$ and $h_2$ the corresponding height of the layers. Using the climatology of ISAS13 atlas and HYCOM re-analysis we make the calculations for four different cases (Tab. C1).

**Table C1.** Characteristics of the layers for the baroclinic cases of the Rossby wave calculation.

| Name | Based on | $h_1$ | $h_2$ | $\rho_1$ | $\rho_2$ |
|------|----------|-------|-------|----------|----------|
| | | [m] | | [kg.m$^{-3}$] | |
| $\mathbf{R_{500}}$ | ISAS13 | 500 | 4500 | 1027 | 1032 |
| $\mathbf{R_{35}}$ | HYCOM | 35 | 4965 | 1025 | 1032 |
| $\mathbf{R_{200}}$ | HYCOM | 200 | 4800 | 1026 | 1032 |
| $\mathbf{R_{600}}$ | HYCOM | 600 | 4400 | 1027 | 1032 |

*Acknowledgements.* Special thanks to the officers and crew of the R/V L'Atalante who operated the OUTPACE cruise (http://dx.doi.org/ 10.17600/15000900). The Argo data were collected and made freely available by the International Argo Project and the national programs

that contribute to it (http://www.argo.ucsd.edu, http://argo.jcommops.org). Argo is a pilot program of the Global Ocean Observing System. The altimeter products were produced by Ssalto/Duacs and distributed by Aviso, with support from CNES (www.aviso.altimetry.fr/duacs/). The HYCOM simulation and re-analysis are sponsored by the National Ocean Partnership Program (hycom.org). Thanks to Luiz Neto for

5    developing a float profile toolbox, Alain Fumenia for the oxygen calibration of the OUTPACE floats, Fabienne Gaillard for ISAS analysis and Xavier Carton for his proof-reading and comments. The authors are grateful for the support of the OUTPACE project (PIs: Thierry Moutin and Sophie Bonnet) funded by the French research national agency (ANR-14-CE01-0007-01), the LEFE-CyBER program (CNRS-INSU), the GOPS program (IRD), the CNES, and from European FEDER Fund under project 1166-39417. We also thank ERC Remocean, the LEFE and MOM programs, as well as the Contrat de Projets Etat - Polynésie française for their financial supports to the THOT project lead by

10   Elodie Martinez (IRD). Simon Barbot thanks the Physics team of MIO and the OUTPACE project for the funding of his internship, as well as the LATEX project (PIs: Anne Petrenko and Frédéric Diaz) for complementary support.



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
