# Peer review of "Intermediate water flows in the South West Pacific: as revealed by individual Argo floats trajectories and a model re-analysis"

_Biogeosciences, 2018_

## Referee Comment (RC1) · Anonymous Referee #1 · 23 Apr 2018

The paper is constructed on float trajectories, dissolved oxygen data and HYCOM model simulations. Two really different topics are discussed with little link between the two. The first one: how eddy structures contribute to water mixing at intermediate depths close to Queensland, and the second one on characterizing wave structures between two intermediate jets in middle tropical Pacific. Little use is made of the float data for this second theme (and even for the first one), except for the trajectories. In particular, I don't understand why T, S, and density of the floats are not used (in complement or to validate HYCOM simulations, granted that this simulation likely assimilates those data). Furthermore, I did not get fully convinced that the plane waves were observed for almost two whole years, or what the criteria used to determine one optimal couple (frequency, wavelength) really select. In some ways, nonetheless, some limita-

tions of the approach used are reported in the discussion section. I was also expecting in this section a discussion (some hints) on processes that could favor the generation of waves at that period and latitude... A strong signal at 900m surprised me a little for a mode estimated with a thermocline at 200m. Could it be local instability around currents at intermediate depth (or further up in water column?), and maybe some frequency selection due to such dynamical processes, and the dispersive/propagative properties of Rossby waves in a horizontally-sheared environment. These are many elements missing that would contribute to make the paper valuable.

The paper also complements to some extent results discussed in earlier papers (Rousselet et al. 2016, 2017), at least for the first topic. Thus, I do not recommend that the paper be published, as constructed. It would probably be more valuable to focus in more depth on the second topic, so that the results provided that would be easier to assess.

Finally, the paper needs to be thoroughly edited.

What follows are comments or suggestions for changes through the manuscript.

p.1, Line 20: 'place' instead of 'replace' p.1, l. 23: replace 'complete' by 'complement' statement, l. 1 of p. 2: statement was not introduced, but reads as a conclusion statement. Should be supported first by references of what can be done. This sentence would then conclude the paragraph. p. 2, L. 2: replace 'they' by 'there' p. 2, l. 13: 'more deoxygeneated'. Better to write 'less oxygenated' (or 'low oxygen event') p.2, l. 14: 'longer branch'... implication is that water mass is 'older' based on its last contact to the surface, thus less ventilated? p. 2, l. 26: 'highlighted' p. 2, l. 33, and p.4, l.1: 'place' instead of 'replace' p. 3, last line: replace 'the studies float' by 'the floats in this study' p. 4, l. 9: replace 'By memory, it begins...' by 'This cycles starts...' p.4, l. 11: 'like those...' by 'such as those...' p. 4, l. 15: 'immersed' by 'deployed' p. 4, l. 16: 'point to them' by 'refer to them'. I think that this sentence should be rewritten

Presentation of the floats on page 4: I got lost, which are the floats that are Arvor

and Argos-located, and which is the float (656, only? Is it a PROVBIO?) that is iridium located . Then, a discussion argues that there is little influence of surface displacement on the intermediate depth current estimates. I thought that this was not negligible for the Argos-located floats, because of longer time spent at the surface. It would be good to explain what assumption is done and further arguments for why this is not an issue for the paper?

p.5, I don't understand the title of section 2.2 p. 5, l.6: why include kz in k (and thus in lambda), as trajectories are horizontal... (actually, zonal) p. 5, l. 19: what is a 'half float wave'? (the terminology is rather vaguely defined in this chapter). How is the estimate made, in an environment which is clearly not mono-chromatic? Some of the presentation might be necessary, but it has long been presented in various papers (for example Flierl, 1981). We seem to be in the case of no 'trapping' of floats in eddies (surface-intensified signature versus drift at 1000-m); thus classical 'linear' approach. Not completely sure that I understand what is thought after. What is probably assumed is that one considers non-dispersive (and mono-chromatic, at least a dominance of one frequency) waves? This is sort of assumed by the approach. Not sure why an inverse model approach is not feasible. Also, hard to see how quantitative is the approach with this J index. Maybe, this could be tested in the model simulations or in simulated fields made up from a superposition of plane waves over a sheared background, for example.

p. 7, l. 8: two indices...' p. 9, l.10: The last sentence is not clear. Is it the part of the trajectory after October 2016, or the whole trajectory. If this is the full trajectory, this sentence can be removed (which is what I guess from what follows). p. 9, l.20: replace 'deoxygenation events' by 'low oxygen'. Deoxygenation refers to something else, and should be replaced throughout the paper by other words. p. 9, l. 22: there seems to be another low oxygen event between D1 and D2 (although less consistent vertically). p. 9, l. 25: this seems rather hypothetical statement (NVJ is four degrees further north). What is the evidence for that in the two papers referred to? (afterwards, I saw Appendix B figure, and figure 5 that provides quite compelling evidence) p. 11,

figure 5: the southward currents in A are not that strong. Focus on A, and not on C2? Question is density in HYCOM comparable to float density at 300m ? (I expect that as the Argo data are assimilated in HYCOM...) p. 11, l. 12-14: I am not sure what this adds. I think that one could remove these two sentences. p. 12, choice of the J-index. Why this metrics instead of other ones (after all, it is a way to normalize errors in simulating meridional trajectories). When summing the two floats, wouldn't it have been better to normalize the respective two with the variance of respective meridional displacements (p. 8). Fig. 7 on page 14 is rather interesting, but somehow I wonder whether the fit is much better for 687 than for 660 (in particular, for the first part, when the two floats are rather close-by)? p. 15, l. 1-4: the comment on striations in HYCOM being different. I don't fully understand the comment. It would suggest that the model is not fully appropriate to provide the circulation context. p. 15, l.12: 'such as on November 13th' p. 15, l.13: replace 'globally' by 'usually' (and then again 'like' by 'such as' on lines 12 and 13) p.15, l. 14: ambiguous sentence. Is the S (density) meridional gradient just near 900m depth, or do you select the value at 900m depth to illustrate the meridional gradient. Again, if HYCOM can be used, it would also be interesting to indicate whether the salinity, density and water mass gradient is indicated with the two floats. Otherwise, the earlier comment on the different striation in HYCOM compared to the observations raises some questions on where the fronts are located.

p. 15, l. 25: 'It cannot be ruled out...' You could write more directly: 'It is likely that this eddy influenced the float trajectory...' p. 15, l. 27: 'during the whole period' p. 17 and 18, first par. Of 4.1 Coral Sea. I am not completely sure on how different the two hypotheses are, and the way they are introduced could be clarified. Again, how different/similar is HYCOM compared to the float data. Do we trust the position of the eddies in HYCOM, and if so to within which uncertainty? p. 18, l. 12: 'of turbulence' p. 18, l. 15: replace 'could' by 'would'. p. 18, l. 26: rewrite the sentence: 'We detail this sentence...' p. 18, l. 31: 'the precision of the values...' I was indeed expecting a discussion on uncertainties on these values, when they were presented. p. 19, l. 14-15. I think that the sentence is not required. There is interest both in the average flow

and in knowing that it is variable. p. 20, l. 5: I am sure that this is not a 'new' proposed pathway, but I don't have a clear example on mind. p. 20, l. 10: 'recourse to' not used properly: not sure what is meant, but maybe 'biogechemical model simulations would help. . .' p. 20, l. 13: 'water mass properties. . .' p. 20, l. 16 'correction the impact of. . .' I am not sure what is corrected? p. 20, l. 29: the sentence could be 'This study also underlined the importance of eddies in addition to the waves for the mesoscale dynamics of intermediate flows'

––––––––––––––––––––––

---

## Referee Comment (RC3) · Anonymous Referee #2 · 18 May 2018

Review of 'Intermediate water flows in the South West Pacific from OUTPACE and THOT Argo floats', by Barbot, Petrenko and Maes

This paper explores aspects of intermediate water flows in the southwest and central Pacific using individual Argo floats. From an analysis of low oxygen intrusions in a float flowing within the NCJ the authors argue that they originate from the NVJ – advected by cyclonic eddies. The variable oscillatory trajectories of zonally propagating floats are examined in detail. The Lagrangian and Eulerian characteristics are determined. Their analysis shows that a single Rossby wave can explain the trajectories of 2 floats travelling in opposite directions. A further section considers the salinity and density structure at 1000m (the Argo parking depth) and the impact on the float trajectories.

I find that the paper lacks any overall focus. It comes across as a few mildly interesting but unconnected observations. They all involve aspects of the circulation in the region but ultimately do not make a coherent story. The content is simply not strong enough to be suitable for publication.

Further Comments

The title is very specialized. How many readers would know the meaning of the OUTPACE and THOT acronyms?

Page 1, Line 16 - …(WTSP)  is of interest to the biogechemical …

Page 2, line 18 – Change to 'BGC data from the float allow us to determine whether it has encountered water masses coming from the NVJ.

2, line 35 - …during the OUTPACE cruise or  the THOT project.

4, line 1 – we also  place the float trajectories ….

4, line 10 -  The float descends to a depth around 1000m,  called …

4, line 25 – An error estimate should be provided.

5, line 3 – Trajectory description for  a wave approach

5, line 4 – What type of waves?

11, line 14 – we  hypothesize that such …

15, line 2 - …we are able to  place the trajectory…

15, line 6 – This appears to be stating the very obvious point about the two different velocity observations.

18, 4 - ..and then be transported southwards  by the current located…

18, line 7 - ..widen the  understanding of the connection …

18, line 7 – and the NCJ and  suggest that there be an explicit consideration of mesoscale  eddy variability in future modeling  approaches.

20, line 8 – …. To replace place the

20, line 13 – of water mass properties,

20, line 16 – form a favorable  environment…

20, line 16 – Not sure what you mean by the sentence beginning 'Correcting the impact of the Lagrangian observation

20, line 20 – would require  a longer toime series.

A large portion of the paper (1/3) deals with the methods and description of decomposing the float trajectory into a wave framework.  A briefer description of the method would be more appropriate for the paper.

The figure generally are poor, the captions are tiny and difficult to read.  Figure 9 is simply impossible to distinguish any of the contours.  To summarize the figures are below the standard required for publication.  Any submission for review should provide figures that are ready for publication.

---

## Author Comment (AC1) · 8 Jun 2018

Response to : Anonymous Referee #1

The paper is constructed on float trajectories, dissolved oxygen data and HYCOM model simulations. Two really different topics are discussed with little link between the two. The first one: how eddy structures contribute to water mixing at intermediate depths close to Queensland, and the second one on characterizing wave structures between two intermediate jets in middle tropical Pacific. Little use is made of the float data for this second theme (and even for the first one), except for the trajectories. In particular, I don't understand why T, S, and density of the floats are not used (in complement or to validate HYCOM simulations, granted that this simulation likely assimilates those data). Furthermore, I did not get fully convinced that the plane waves were observed for almost two whole years, or what the criteria used to determine one optimal couple (frequency, wavelength) really select. In some ways, nonetheless, some limitations of the approach used are reported in the discussion section. I was also expecting in this section a discussion (some hints) on processes that could favor the generation of waves at that period and latitude. . . A strong signal at 900m surprised me a little for a mode estimated with a thermocline at 200m. Could it be local instability around currents at intermediate depth (or further up in water column?), and maybe some frequency selection due to such dynamical processes, and the dispersive/propagative properties of Rossby waves in a horizontally-sheared environment. These are many elements missing that would contribute to make the paper valuable.

The paper also complements to some extent results discussed in earlier papers (Rousselet et al. 2016, 2017), at least for the first topic. Thus, I do not recommend that the paper be published, as constructed. It would probably be more valuable to focus in more depth on the second topic, so that the results provided that would be easier to assess.

As quoted by Moutin et al. (2017, preface of the special issue), "the goal of this special issue is to present the knowledge obtained concerning the functioning of WTSP ecosystems and associated biogeochemical cycles based on the datasets acquired during the OUTPACE experiment". It takes place along a 4000km zonal transect from the north of New Caledonia to the French Polynesia. As it is well known, the spatio-temporal domain of an oceanographic cruise is characterized by horizontal stirring generated by ocean circulation at the mesoscale, inducing strong variability in different parameters. Consequently, studying the relevant large-scale dynamics and its potential links to the ephemeral and local gradients due to mesoscale activity is essential to provide a broad overview of the observations, from the time of the cruise and beyond. In this long-term and large scale context, it is true that various physical processes could interplay. It is, in some way, the objectives of our study to document and to explain the processes that favor the generation of the observed data collected from a Lagrangian point of view (i.e., Argo floats). We agree that some elements were previously missing. We have taken care, with great attention, of the proposed suggestions in order to make a valuable manuscript, and hence have included references to earlier papers in both regions under consideration and additions in the proposed methodology. As the main results of the OUTPACE cruise are based on the full transect, and especially on the zonal large-scale gradient of biology and biogeochemistry of diazotroph organisms, we continue to believe that a dynamical study of the different regimes in the intermediate waters is highly relevant for this special issue. , We have also taken into account the editorial propositions and the very constructive suggestions on our approach. We sincerely thank the reviewer for his/her patient and careful reading. Details of our responses and additions to our original manuscript are described below.

Finally, the paper needs to be thoroughly edited. What follows are comments or suggestions for changes through the manuscript.

p.1, Line 20: 'place' instead of 'replace' Done

p.1, l. 23: replace 'complete' by 'complement' Done

statement, l. 1 of p. 2: statement was not introduced, but reads as a conclusion statement. Should be supported first by references of what can be done. This sentence would then conclude the paragraph. Done, the paragraph has become :

"Autonomous Argo floats are profilers parked at 1000 m depth and reaching the surface every 10 days, where they transmit their measurements and location (further information on the cycle of the different Argo floats we used is detailed in the next section).

Current speeds at the parking depth of the floats are then calculated, leading to horizontal gridded maps of velocities either global (Davis, 2005, Ollitrault and Rannou, 2013) or regional (Cravatte et al., 2012). Other approaches are developed in order to study the intermediate circulation, for example by taking into account the spreading of different float trajectories from a same position (Sevellec et al., 2017). Moreover, when Argo floats are BCG, on top of allowing the study of intermediate circulation, they provide measurements of biogeochemistry and biology parameters over the first thousand meters of the water column."

p. 2, L. 2: replace 'they' by 'there' Done

p. 2, l.13:'more deoxygeneated'. Better to write 'less oxygenated' (or 'low oxygen event') Done

p.2, l.14: 'longer branch'. . . implication is that water mass is 'older' based on its last contact to the surface, thus less ventilated? Longer 'path without ventilation' is more accurate, the sentence has been corrected and now is:

"The NVJ is older, in the sense that it originates from a longer path than the NCJ, since its last contact to the surface. Hence, without ventilation, the DOXY of the NVJ intermediate waters is lower than the NCJ DOXY.

p. 2, l. 26: 'highlighted' Done

p. 2, l. 33, and p.4, l.1:'place' instead of 'replace' Done

p. 3, last line: replace 'the studies float' by 'the floats in this study' Done

p. 4, l. 9: replace 'By memory, it begins. . .' by 'This cycles starts. . .' Done

p.4, l. 11:'like those. . .' by 'such as those. . .' Done

p. 4, l. 15: 'immersed' by 'deployed' Done

p. 4, l. 16:'point to them' by 'refer to them'. I think that this sentence should be rewritten Done, the sentence has become :

"Hereafter, we only use the three last digits of the float number to refer to them, i.e. float 656 refers to #6901656.

Presentation of the floats on page 4: I got lost, which are the floats that are Arvor and Argos-located, and which is the float (656, only? Is it a PROVBIO?) that is iridium located . To clarify that we have added the sentence :

"In this study, we use three PROVBIO floats using Iridium (656, 660 and 687) and two ARVOR floats using ARGOS (671 and 679)."

Then, a discussion argues that there is little influence of surface displacement on the intermediate depth current estimates. I thought that this was not negligible for the Argos-located floats, because of longer time spent at the surface. It would be good to explain what assumption is done and further arguments for why this is not an issue for the paper? It is true that ARVOR floats (due to longer times remaining at the surface for telecommunications) have greater sensitivity to surface currents than PROVBIO ones; see the table below that has been added to the article.

Table 1: Statistics for the different floats between surface and deep trajectories, the results are presented as 'mean $\pm$ std'.

| Float | Type | Surface distance [km] | Deep distance [km] | Surface speed [cm/s] | Deep speed [cm/s] |
|---|---|---|---|---|---|
| 656 | PROVBIO | $0.44 \pm 0.23$ | $17.32 \pm 18.18$ | $29.57 \pm 15.83$ | $4.30 \pm 3.93$ |
| 660 | PROVBIO | $0.35 \pm 0.18$ | $15.96 \pm 9.28$ | $25.77 \pm 13.16$ | $3.65 \pm 1.87$ |
| 671 | ARVOR | $6.08 \pm 3.44$ | $25.30 \pm 14.04$ | $28.37 \pm 16.12$ | $3.00 \pm 1.67$ |
| 679 | ARVOR | $4.84 \pm 2.38$ | $21.88 \pm 12.17$ | $22.44 \pm 11.07$ | $2.60 \pm 1.45$ |
| 687 | PROVBIO | $0.33 \pm 0.19$ | $8.54 \pm 6.53$ | $24.26 \pm 13.20$ | $3.66 \pm 1.89$ |

We have also replaced the sentence "After some verifications...deep displacement." by the paragraph :

"Table 1 shows the mean properties of displacements for each studied float. First, it highlights the differences in surface distances between PROVBIO and ARVOR floats. At the surface, ARVOR floats drift over a distance about 10 times greater than PROVBIO floats. This is due to the longer time they spent at the surface (6h for ARVOR floats and 24min for PROVBIOs). The ratio between deep and surface distances is a factor of 30 for PROVBIO floats and still 4 for ARVOR ones. Float 656 exhibits anomalously high standard deviation for its deep distance and deep speed. These high values, of the same order as the mean ones, are due to the period during which the float was grounded on the sea floor in the Queensland plateau. Otherwise we have concluded that surface displacements can be neglected compared to deep displacements without doubt for PROVBIO floats and with caution for ARVOR floats. Hence, when needed in the wave section, we will only use the former ones and consider that the trajectory dynamics are mainly due to deep circulation processes. The discussion of such considerations is thoroughly made by Ollitrault and Rannou (2013)."

p.5, I don't understand the title of section 2.2 The idea was to consider the trajectory as wave signature ("wave approach"). It should be clearer that way : "Wave characteristics from float trajectories"

p. 5, l.6: why include $k_z$ in $k$ (and thus in lambda), as trajectories are horizontal. . . (actually, zonal) We were explaining the decomposition in a very general case. We have suppressed this part and directly use « $k = k_{lon}$ » for a zonal case.

p. 5, l. 19: what is a 'half float wave'? (the terminology is rather vaguely defined in this chapter). We have replaced the sentence "we choose to determine...wave ones." by :

"we choose not to describe the complete float wave characteristics $(T, \lambda)$ but their halves $(T/2, \lambda/2)$. In practice, this means we measure time and distance from crest to trough (and so on) rather than from crest to crest, for each studied float (floats 660, 671, 679 and 687, Fig.2). Hence, it allows to have more estimates when incomplete cycles are present." The estimations of $\lambda/2$ are made from the position maps and the ones of $T/2$ from the float time series of latitude.

How is the estimate made, in an environment which is clearly not mono-chromatic? As explained in the text (p.5 l.14 of the first submitted version), because of the shortness of the time series with regard to the sampling frequency, we cannot perform a Fourier transform or a wavelet analysis. Hence we decided to try to describe the wave in the simplest way as a mono-chromatic wave.

Some of the presentation might be necessary, but it has long been presented in various papers (for example Flierl, 1981). We seem to be in the case of no 'trapping' of floats in eddies (surface-intensified signature versus drift at 1000-m); thus classical 'linear' approach. Since this article is proposed for publication in the OUTPACE Special Issue mainly composed of biogeochemistry articles, we made the choice to start the explanation of the methods with a very generic/basic framework without using the complete terminology used by Flierl (1981), which is harder to comprehend for non physical oceanographers. Nonetheless, we have added the reference to Flierl (1981) as well as part of his terminology. The first paragraph of section 2.2 has become :

"Here the objective is to find the characteristics of a single wave that could explain the float trajectories which represent both retrograde and prograde circulation (retrograde when Lagrangian motion is in the opposite direction as the wave propagation ; prograde when Lagrangian motion and wave propagation are in the same direction; as defined in Flierl, 1981). We have also added some diagnostics to characterize the best fitting wave. We have added a paragraph at the end of the section 2.2 :

"According to Flierl (1981), for each float, we can define

$$\varepsilon = \frac{u_0}{c} \tag{1}$$

where $c$ is the phase speed of the wave ($c = \frac{\omega}{k}$). The sign of $\varepsilon$ indicates the type of motion of the float (prograde or retrograde) and its amplitude compares the particle velocity to the wave speed." We have also added the results of this calculation at the end of Section 3.2.1. As mentioned in the results section (p.15 l.24 to 28 of the first submitted version) , we observed eddy structures in the velocity field of HYCOM but this hypothesis does not explain the entire 660 trajectory (only its beginning is influenced by an eddy). Moreover no eddy is present along the entire 687 trajectory. Furthermore, the dominance of deep distances over surface ones (see table above) does not induce to decouple surface signature versus deep drift. Indeed the trajectories of the floats are not shifted only during the surface periods of their cycles but mostly during their parking depth periods. Added to the general absence of eddies, we consider that all this suggests that the wave signature is present at this deep depth.

Not completely sure that I understand what is thought after. What is probably assumed is that one considers non-dispersive (and mono-chromatic, at least a dominance of one frequency) waves? This is sort of assumed by the approach. Yes, it is. That is why we had used the term "plane wave". We have tried to make it clearer substituting "Because of the zonal tendency...system of equations" by : "Because of the zonal tendency of the studied float trajectories (Fig. 2), we express a simple case of the current perturbations due to a homogeneous, monochromatic wave (hereafter plane wave propagation) with the following system of equations..."

Not sure why an inverse model approach is not feasible. It is not the choice that we made; but we have added it as another perspective in the last section : "Another approach could also be made with an inverse model."

Also, hard to see how quantitative is the approach with this J index. Maybe, this could be tested in the model simulations or in simulated fields made up from a superposition of plane waves over a sheared background, for example. Our approach is to test the hypothesis that one single wave could be responsible for these float trajectories. The approach of the J index is to get a first guess of what could be the characteristics of such wave. This could be confirmed/denied with a more complex study, inverse methods or study implying several waves/eddies. The improvement of the calculation and representation of J-index is described below.

p. 7, l. 8: two indices. . .' Done

p. 9, l.10: The last sentence is not clear. Is it the part of the trajectory after October 2016, or the whole trajectory. If this is the full trajectory, this sentence can be removed (which is what I guess from what follows). Indeed it is the full trajectory, we removed the last sentence.

p. 9, l.20: replace 'deoxygenation events' by 'low oxygen'. Deoxygenation refers to something else, and should be replaced throughout the paper by other words. Thank you for also pointing that to us; we did substitute D1 and D2 by O1 and O2.

p. 9, l. 22: there seems to be another low oxygen event between D1 and D2 (although less consistent vertically). Yes, but we chose to focus on the events which are below 140 $\mu$mol/kg (p.9 l.20) because, in this way, we can avoid other hypotheses explaining low DOXY (as strong primary production for example).

p. 9, l. 25: this seems rather hypothetical statement (NVJ is four degrees further north). What is the evidence for that in the two papers referred to? (afterwards, I saw Appendix B figure, and figure 5 that provides quite compelling evidence) The two papers refer to the NVJ intrusion on the NCJ pathway hypothesis ; we have added a reference to Figure 5b to justify this statement.

p. 11,figure 5: the southward currents in A are not that strong. Focus on A, and not on C2? We did focus on C2 : "Figure 5b clearly shows that the NVJ waters can be associated with C2 for instance." (p.11 l.11 of the first submitted version and p.11 l.21 of the revised version that will be transmitted as soon as requested)

Question is density in HYCOM comparable to float density at 300m ? (I expect that as the Argo data are assimilated in HYCOM. . .) We are confident in the HYCOM results as the NCODA system (Cummings, 2005; Cummings and Smedstad, 2014), that is used by the HYCOM model, assimilates available satellite altimeter observations, satellite and in situ SST as well as vertical temperature and salinity profiles (from XBTs, Argo floats and moored buoys). It should be noticed that the profiles from our Argo floats are flagged good or probably good, meaning that these data are likely considered by the re-analysis. A comparison of Argo data with HYCOM re-analysis is shown in Fig.C1,C2 and C3 (next pages). Since a detailed validation is not the purpose of the study, these figures have been added in an appendix of the article, and not in the main text. We add the following paragraph at the beginning of the Section 3.1.1 :

"In addition, we also expose the corresponding Argo profiles from HYCOM re-analyses in the Appendix C. As expected from a re-analyses that assimilates different measurements including Argo float data, the salinity and temperature are in good agreement with the float measurements."

p. 11, l. 12-14: I am not sure what this adds. I think that one could remove these two sentences. This was to exhibit the case of D1/O1 but we agree that it does not add much to the article; we have removed the sentences.

p. 12, choice of the J-index. Why this metrics instead of other ones (after all, it is a way to normalize errors in simulating meridional trajectories). When summing the two floats, wouldn't it have been better to normalize the respective two with the variance of respective meridional displacements (p. 8). Thanks for this proposition. At first, we thought that this would also help to get a fit closer to the 660 trajectory. Nonetheless, it turns out that the normalization does not change the results (Figure J below). To show the J index quantitatively, we have changed the red dots of Figure 6B of the first submitted version, with the real value of J and hence also transformed the colorbar to its right to represent the normalized J scale (see below).

Fig. 7 on page 14 is rather interesting, but somehow I wonder whether the fit is much better for 687 than for 660 (in particular, for the first part, when the two floats are rather close-by)? As explained at the end of the results (p.15 l.24), the beginning of the 660 trajectory is influenced by an eddy and otherwise the rest of the trajectory is also more irregular than the 687 trajectory. The first perspective that we mention in the discussion (to consider a variation of the zonal background current) would probably allow to get a better fit (Fig.10).

p. 15, l. 1-4: the comment on striations in HYCOM being different. I don't fully understand the comment. It would suggest that the model is not fully appropriate to provide the circulation context. This was a simple observation. This difference could be due to the fact that HYCOM re-analysis figures only represent an instantaneous snapshot versus a more than 10-year mean for the figure of Ollitrault and Colin de Verdière (2014). A more complete study should compare the two approaches and their impact on striations, but it is not the purpose of our study and needs further investigations.

p. 15, l.12: 'such as on November 13th' Done

p. 15, l.13: replace 'globally' by 'usually' (and then again 'like' by 'such as' on lines 12 and 13) Done

p.15, l. 14: ambiguous sentence. Is the S (density) meridional gradient just near 900m depth, or do you select the value at 900m depth to illustrate the meridional gradient. Figure 9a shows a meridional gradient of salinity between 1200 m and 500 m depth, but the gradient that is being used in our study is the one at the parking depth of the float, so around 1000 m depth. To clarify, we have substituted the sentence "Meridional cross-sections...around 900 m" by :

"Meridional cross-sections of HYCOM re-analysis (Fig. 9a) show a salinity gradient in the intermediate waters, between 500 m and

[Figure]

FIGURE C1 : Profiles of (left side) float 656 and (right side) corresponding data from HYCOM re-analysis over depth and time for (a,e) absolute salinity, (b,f) conservative temperature and (c,g) density. Every colored point corresponds to a measurement. The black lines indicate the isopycnals from 1024 to 1031 kg.m$^{-3}$.

[Figure]

FIGURE C2 : Profiles of (left side) float 660 and (right side) corresponding data from HYCOM re-analysis over depth and time for (a,e) absolute salinity, (b,f) conservative temperature and (c,g) density. Every colored point corresponds to a measurement. The black lines indicate the isopycnals from 1024 to 1031 kg.m$^{-3}$.

[Figure]

FIGURE C3 - Profiles of (left side) float 687 and (right side) corresponding data from HYCOM re-analysis over depth and time for (a,e) absolute salinity, (b,f) conservative temperature and (c,g) density. Every colored point corresponds to a measurement. The black lines indicate the isopycnals from 1024 to 1031 kg.m$^{-3}$.

[Figure]

FIGURE J : Detail of J-index calculation for (left) the float 660, (center) the float 687 and (right) the overall J-index in a case without normalization (upper panels) and normalized with the standard deviation of the latitude of each float.

[Figure]

FIGURE 6 : Difference between observed Lagrangian float wave characteristics and Lagrangian theoretical wave ones (index **I**, left colorbar) for (a) the sum of floats 660 and 687 and (b) the associated minimum calculations (normalized index **J**, right colorbar) with regard to dispersion equations of Rossby wave (solid line), Kelvin wave (dotted line), Kevin-Helmholtz instability wave (break line). Subscript b means a barotropic case, subscripts with number means a baroclinic case with a stratification at the depth of this number. The negative wavelength describes a westward wave. Black dots indicate the observed Lagrangian parameters; note that on (b) only the 687 one is inside the range; the darker red dots show the minima of the index hence the best parameters that can explain both trajectories; the black surrounded by white dot is the couple ($\omega_{TMe}$ , $k_{TMe}$) that better fits the 660 and 687 float trajectories.

1200 m, around the AAIW. We can note the meridional gradient of salinity at the parking depth of the float, around 1000m"

Again, if HYCOM can be used, it would also be interesting to indicate whether the salinity, density and water mass gradient is indicated with the two floats. This important point has been raised previously and, please, refer to figures C2 and C3 above.

Otherwise, the earlier comment on the different striation in HYCOM compared to the observations raises some questions on where the fronts are located. Already answered; indeed we can not compare them directly like that.

p. 15, l. 25: 'It cannot be ruled out. . .' You could write more directly: 'It is likely that this eddy influenced the float trajectory. . .' Done

p. 15, l. 27: 'during the whole period' Done

p.17 and 18, first par. Of 4.1 Coral Sea. I am not completely sure on how different the two hypotheses are, and the way they are introduced could be clarified. The first one corresponds to a transport inside the core of a mesoscale structure (here C2) ; the second one to a transport on the edges of structures (here between C2 and A). We have substituted the sentence "Hence, we can also make a second...C2 to the west." by :

"Hence, we propose a second hypothesis of an edge transport for the NVJ waters going south. These waters could be first carried by the northern edge of A and then be transported southward thanks to the current located between A to the east and C2 to the west."

Again, how different/similar is HYCOM compared to the float data. Do we trust the position of the eddies in HYCOM, and if so to within which uncertainty? As explained above, HYCOM assimilates Argo float data. We do trust the position of C2 and A in HYCOM re-analysis because we also observe them in AVISO velocity maps. This is not the case for C1 which position differs between AVISO and HYCOM re-analysis. But again, to quantify the uncertainty of HYCOM re-analysis compared to observations would require a complete study and is beyond the scope of our study.

p. 18, l. 12: 'of turbulence' Done

p. 18, l. 15: replace 'could' by 'would'. Done

p. 18, l. 26: rewrite the sentence: 'We detail this sentence. . .' Done

p. 18, l. 31: 'the precision of the values. . .' I was indeed expecting a discussion on uncertainties on these values, when they were presented. Because of the complex shape of the J-index values (see above) we cannot give a precision like $\pm$ XX m-1. On the figure we can observe that the minima are 'organized' by lines, which is due to the conversion from Lagrangian wave couple characteristics to Eulerian ones. Indeed, many Eulerian couples could refer to a single Lagrangian couple if all the characteristics were variable. Unfortunately, we do not have any information on a potential Eulerian wave characteristic, especially at that depth (no mooring data in the region and period). As explained by Flierl (1981), "the Eulerian spectrum may contain only one frequency while floats set at different location will have different periods.". Thus, we constrained our problem using two floats with distinct Lagrangian wave characteristics to find a single Eulerian couple of characteristics.

p. 19, l. 14-15. I think that the sentence is not required. There is interest both in the average flow and in knowing that it is variable. Done

p. 20, l. 5: I am sure that this is not a 'new' proposed pathway, but I don't have a clear example on mind. We have changed the sentence "Moreover, the hypothesis...far as we know." by :

"Moreover, the hypothesis that the waters can be transported on the edge of several eddies is a new proposed pathway for NVJ waters."

p. 20, l. 10: 'recourse to' not used properly: not sure what is meant, but maybe 'biogechemical model simulations would help. . .' « resort to » should be better Done

p. 20, l. 13: 'water mass properties. . .' Done

p. 20, l. 16 'correction the impact of. . .' I am not sure what is corrected? We have changed the sentence "Correcting the impact...zonal directions." by :

"In order to convert Lagrangian wave observations to Eulerian ones in a simplified case, we concentrated on two floats, heading in opposite zonal directions. Moreover, we chose two PROVBIO floats in order to assure that their motions are representative of depth dynamics (as explained in Section 2.1)."

p. 20, l. 29: the sentence could be 'This study also underlined the importance of eddies in addition to the waves for the mesoscale dynamics of intermediate flows' Done

---

## Author Comment (AC2) · 8 Jun 2018

Response to : Anonymous Referee #2

This paper explores aspects of intermediate water flows in the southwest and central Pacific using individual Argo floats. From an analysis of low oxygen intrusions in a float flowing within the NCJ the authors argue that they originate from the NVJ–advected

by cyclonic eddies. The variable oscillatory trajectories of zonally propagating floats are examined in detail. The Lagrangian and Eulerian characteristics are determined. Their analysis shows that a single Rossby wave can explain the trajectories of 2 floats travelling in opposite directions. A further section considers the salinity and density structure at 1000m (the Argo parking depth) and the impact on the float trajectories.

I find that the paper lacks any overall focus. It comes across as a few mildly interesting but unconnected observations. They all involve aspects of the circulation in the region but ultimately do not make a coherent story. The content is simply not strong enough to be suitable for publication.

Our manuscript contains two parts that have been found interesting by specialists of the Coral sea concerning the first part, and more generally by physical oceanographers concerning the wave impact at intermediate depth deduced from Argo trajectory for the second one. Moreover, they both have repercussions on biogeochemistry that are of interest to a more expanded community, and in particular to the OUTPACE community; hence the submission as one paper in this Special Issue. Another reason for gathering them in one manuscript is that both studies derive from focusing on float trajectories taken on individual basis; point that we propose to stress with the new title (see below).

Further Comments

The title is very specialized. How many readers would know the meaning of the OUTPACE and THOT acronyms? Another proposition is: "Intermediate water flows in the South West Pacific : contribution from individual Argo floats"

Page 1, Line 16 - ...(WTSP) interests is of interest to the biogechemical ... Done

Page 2, line 18 – Change to 'BGC data from the float allow us to determine whether it has encountered water masses coming from the NVJ. Done

2, line 35 - ...during the OUTPACE cruise or in the framework of the THOT project. Why are you canceling the signification of the THOT acronym and the reference to THOT ? "(TaHitian Ocean Time series, Martinez et al. 2015)"

4, line 1 – we also replace place the float trajectories .... Done here and everywhere else in the article.

4, line 10 - For memory, it begins with the descent of the float The float descends to a depth around 1000m, called ... We have modified the sentence accordingly : "This cycle starts with the descent of the float to a depth around 1000 m, called..."

4, line 25 – An error estimate should be provided. We have added the Table:

**Table 1.** Statistics for the different floats between surface and deep trajectories, the results are presented as 'mean ± std'.

| Float | Type | Surface distance [km] | Deep distance [km] | Surface speed [cm/s] | Deep speed [cm/s] |
|-------|------|-----------------------|---------------------|----------------------|-------------------|
| 656 | PROVBIO | $0.44 \pm 0.23$ | $17.32 \pm 18.18$ | $29.57 \pm 15.83$ | $4.30 \pm 3.93$ |
| 660 | PROVBIO | $0.35 \pm 0.18$ | $15.96 \pm 9.28$ | $25.77 \pm 13.16$ | $3.65 \pm 1.87$ |
| 671 | ARVOR | $6.08 \pm 3.44$ | $25.30 \pm 14.04$ | $28.37 \pm 16.12$ | $3.00 \pm 1.67$ |
| 679 | ARVOR | $4.84 \pm 2.38$ | $21.88 \pm 12.17$ | $22.44 \pm 11.07$ | $2.60 \pm 1.45$ |
| 687 | PROVBIO | $0.33 \pm 0.19$ | $8.54 \pm 6.53$ | $24.26 \pm 13.20$ | $3.66 \pm 1.89$ |

To satisfy both you and the other reviewer, we have also replaced the sentence "After some verifications...deep displacement." by the paragraph :
"Table 1 shows the mean properties of each studied float. First, it highlights the differences in surface distances between PROVBIO and ARVOR floats. At the surface, ARVOR floats drift over a distance about 10 times greater than PROVBIO floats. This

is due to the longer time they spent at the surface (6h for ARVOR floats and 24min for PROVBIO one). The ratio between deep and surface distances is a factor of 30 for PROVBIO floats and still 4 for ARVOR ones. Float 656 exhibits anomalously high standard deviation for its deep distance and deep speed. These high values, equal to the mean ones, are due to the period during which the float was grounded on the sea floor in the Queensland plateau. Otherwise we have concluded that surface displacements can be neglected compared to deep displacements without doubt for PROVBIO floats and with caution for ARVOR floats. Hence, when needed in the wave section, we will only use the former ones and consider that the trajectory dynamics are mainly due to deep circulation processes. The discussion of such considerations is thoroughly made by Ollitrault and Rannou (2013)."

5, line 3 – Trajectory description for the a wave approach We have changed it to : "Wave characteristics from float trajectory"

5, line 4 – What type of waves? We have modified the first sentence of this section to make it more explicit: "Here the objective is to find the characteristics of a single wave that could explain the float trajectories which represent both retrograde and prograde circulation (retrograde when Lagrangian motion is in the opposite direction as the wave propagation; prograde when Lagrangian motion and wave propagation are in the same direction; as defined in Flierl, 1981)".

11, line 14 – we could hypothesized hypothesize that such ... Done

15, line 2 - ...we are able to replace place the trajectory... Done

15, line 6 – This appears to be stating the very obvious point about the two different velocity observations. Sentence removed

18, 4 - ..and then be transported southwards thanks to by the current located... Done

18, line 7 - ..widen the comprehension understanding of the connection ... Done

18, line 7 – and the NCJ and claim for the suggest that there be an explicit consideration

of mesoscale eddies eddy variability in future modeling approach approaches. Done

20, line 8 – .... To replace place the Done

20, line 13 – of water masses properties, Done

20, line 16 – form a favorable context environment... Done

20, line 16 – Not sure what you mean by the sentence beginning 'Correcting the impact of the Lagrangian observation We have changed the sentence "Correcting the impact...zonal directions." by :
"In order to convert Lagrangian wave observations into Eulerian ones in a simplified case, we concentrated on two floats, heading in opposite zonal directions, and hence providing both cases of prograde and retrograde motions."

20, line 20 – would require to wait until the time series are longer a longer time series. Done

A large portion of the paper (1/3) deals with the methods and description of decomposing the float trajectory into a wave framework. A briefer description of the method would be more appropriate for the paper. We agree that the method section is a bit large but, since this article is proposed for publication in the OUTPACE Special Issue mainly composed of biogeochemistry articles, we made the choice to start the explanation of the methods with a very generic/basic framework, accessible to all oceanographers and not only physical oceanographers.

The figure generally are poor, the captions are tiny and difficult to read. Figure 9 is simply impossible to distinguish any of the contours. To summarize the figures are below the standard required for publication. Any submission for review should provide figures that are ready for publication., all the figure were reprocess to get a decent resolution Done. We have reprocessed all the figures and set them with an adequate resolution to avoid any inconvenient or misreading. The revised version can be transmitted as soon as requested.